# Exploring the Complexities of Seafood: From Benefits to Contaminants

**DOI:** 10.3390/foods14091461

**Published:** 2025-04-23

**Authors:** Bettina Taylor, Kelvin Fynn Ofori, Ali Parsaeimehr, Gulsun Akdemir Evrendilek, Tahera Attarwala, Gulnihal Ozbay

**Affiliations:** 1Human Ecology Department, Delaware State University, Dover, DE 19901, USA; bctaylor@desu.edu; 2Integrative PhD Program in Agriculture, Food and Environmental Sciences, College of Agriculture, Science and Technology, Delaware State University, Dover, DE 19901, USA; kfofori22@students.desu.edu; 3Department of Agriculture and Natural Resources, Delaware State University, Dover, DE 19901, USA; ali.parsaeimehr@sdstate.edu (A.P.); tattarwala17@students.desu.edu (T.A.); 4Cooperative Extension, University of Maine, Orono, ME 04469, USA; gulsun.akdemir@maine.edu

**Keywords:** seafood, nutritional benefits, foodborne diseases, food safety

## Abstract

Seafood plays a vital role in human diets worldwide, serving as an important source of high-quality protein, omega-3 fatty acids, and essential vitamins and minerals that promote health and prevent various chronic conditions. The health benefits of seafood consumption are well documented, including a reduced risk of cardiovascular diseases, improved cognitive function, and anti-inflammatory effects. However, the safety of seafood is compromised by multiple hazards that can pose significant health risks. Pathogenic microorganisms, including bacteria, viruses, and parasites, in addition to microbial metabolites, are prominent causes of the foodborne diseases linked to seafood consumption, necessitating reliable detection and monitoring systems. Molecular biology and digital techniques have emerged as essential tools for the rapid and accurate identification of these foodborne pathogens, enhancing seafood safety protocols. Additionally, the presence of chemical contaminants such as heavy metals (e.g., mercury and lead), microplastics, and per- and polyfluoroalkyl substances (PFASs) in seafood is of increasing concern due to their potential to accumulate in the food chain and adversely affect human health. The biogenic amines formed during the microbial degradation of the proteins and allergens present in certain seafood species also contribute to food safety challenges. This review aims to address the nutritional value and health-promoting effects of seafood while exploring the multifaceted risks associated with microbial contamination, chemical pollutants, and naturally occurring substances. Emphasis is placed on enhanced surveillance, seafood traceability, sustainable aquaculture practices, and regulatory harmonization as effective strategies for controlling the risks associated with seafood consumption and thereby contributing to a safer seafood supply chain.

## 1. Introduction

Seafood is a vital component of global nutrition and economic stability, offering a rich source of high-quality protein, omega-3 fatty acids, and essential micronutrients such as iodine, selenium, and vitamins A and D [1]. Celebrated for its numerous health benefits, seafood consumption is linked to a reduced risk of cardiovascular disease, improved neurological development, and enhanced overall well-being [2]. As the global population grows, the demand for seafood continues to rise, underscoring its critical role in addressing food security. However, the rapid expansion of the seafood industry presents significant challenges that require a deeper understanding of its environmental, biological, and health-related complexities. Advancements in aquaculture technologies, improved distribution networks, and heightened consumer awareness of seafood’s nutritional benefits have fueled the industry’s growth [2]. While these developments contribute to economic and dietary gains, they also raise pressing concerns, such as overfishing, marine habitat degradation, and threats to the sustainability of marine ecosystems. Balancing the increasing demand for seafood with ecological conservation and sustainable resource management is crucial to ensuring the long-term viability of this global industry [1]. Seafood consumption delivers a wealth of scientifically backed health benefits, primarily through its omega-3 fatty acids, particularly eicosapentaenoic acid (EPA) and docosahexaenoic acid (DHA), which support brain development and cardiovascular health [2]. However, the health impacts of these nutrients can vary depending on the seafood type and preparation methods, necessitating a nuanced approach to dietary recommendations. Despite its nutritional advantages, seafood consumption carries potential risks due to contaminants [3]. Heavy metals such as mercury, cadmium, and lead accumulate in the marine food web, especially in predatory fish, posing significant neurotoxic and carcinogenic risks [4]. Emerging contaminants, including microplastics and antimicrobial-resistant bacteria, further complicate seafood safety and demand targeted scientific investigation and mitigation strategies. Microplastics have become a pervasive environmental pollutant, infiltrating aquatic ecosystems and bioaccumulating in fish and shellfish. These particles may carry harmful chemical additives and pathogens, disrupting marine physiology and food web dynamics. When consumed by humans, microplastics act as carriers for toxic compounds, potentially causing inflammation, oxidative stress, and other adverse health effects [5]. Seafood is also a transmitter for foodborne illnesses caused by pathogens such as *Vibrio*, *Salmonella*, and *Listeria*. The growing prevalence of antimicrobial resistance among these pathogens intensifies public health concerns, complicating treatments and increasing the risk of severe infections [6].

Addressing these challenges calls for integrated monitoring and management strategies to minimize bacterial contamination in seafood production and processing. Innovations in molecular biology have revolutionized the detection and analysis of foodborne pathogens in seafood. Techniques such as polymerase chain reaction (PCR), quantitative PCR (qPCR), and next-generation sequencing (NGS) enable the rapid, precise identification of microbial contaminants, playing a crucial role in ensuring seafood safety and quality while meeting regulatory standards [7,8]. This study aims to provide a comprehensive and scientifically grounded examination of the benefits and risks associated with seafood consumption. By addressing global production trends, nutritional benefits, contamination risks, and advancements in pathogen detection, this research underscores the need for interdisciplinary approaches to tackle the environmental, economic, and public health challenges facing the seafood industry. The insights gained will contribute to the development of sustainable seafood practices and informed public health policies.

## 2. Global Demand, Production, and Consumption of Seafood

Seafood includes the fish and aquatic species sourced from fisheries and aquaculture in both freshwater and marine environments and accounts for 17% of the global production of edible meat [1,9]. Globally, the forecasted rapid increase in the human population has called for the sustainable production of foods. As the global food demand continues to rise, there have been suggestions to expand land use for increased food production. However, there are concerns regarding the impact of land expansion on sustainability with climate change and biodiversity [1]. Attention has turned to seafood as a potential solution for increasing the food supply, with a global focus on sustainable fisheries and marine aquaculture [10]. Dietary shifts, an increasing population, demographic and income growth, cultural preferences, and appreciation for the vast nutritional and health benefits of seafood have synergistically been the drivers for the globally rising demand for seafood in recent years [11]. The estimated growth in fish consumption exceeded the growth in the consumption of other animal proteins such as dairy and meat in 2018, resulting in a 62.4% increase in the aggregate volume of fish demand (live weight) within a twenty-year period from 1998 to 2018 [12,13]. Additionally, seafood production in 2018 contributed to over 20% of the global demand for animal proteins and provided employment to more than 108 million people [12]. In developing countries, seafood serves as a good and cheaper choice of dietary protein, micronutrients, and essential fatty acids, and its production from aquaculture and capture fisheries contributes greatly to employment and income among the population [10].

Seafood is the most traded food commodity on the international market, surpassing all other sources of animal proteins [14]. To address the growing demand for seafood, there has been an increased focus on sustainable aquaculture to supplement wild fisheries. While many countries rely on imports to compensate for shortfalls in domestic seafood production, climate change and rapid population growth threaten the sustainability of imports in the future [15]. Although the International Council for the Exploration of the Sea (ICES) member nations have been global leaders in seafood production, inadequate policies governing aquaculture could result in a projected domestic seafood deficit of 7 million tonnes by 2050 [15]. In Africa, aquaculture production has experienced significant growth over the past decades. The production of farmed aquatic animal species increased from 399,622 tonnes in 2000 to 2,316,825 t in 2022, demonstrating an impressive overall growth of 479.8% and an average annual growth rate of 8.3% from 2000 to 2022. From 2020 to 2022, the aquaculture production of farmed aquatic animals in the region increased marginally by only 2.2% (50,500 t) [16]. Countries such as Egypt, Nigeria, Ghana, and Uganda have been key players in boosting seafood production through aquaculture in Africa. This growth can be attributed to capacity building, improved access to facilities, advancements in research and development, foreign support, the promotion of private sector involvement in the aquaculture industry, and the implementation of special programs by the Food and Agriculture Organization (FAO) and other agencies [17]. Asia leads the global aquaculture production of aquatic animals, contributing approximately 88% (83,399,000 t, live weight equivalent) of the global total in 2022. It is followed by the Americas (4,958,000 t) and Europe (3,503,000 t), while Oceania recorded the lowest output at 235,000 tonnes [16] (Figure 1a).

Global seafood production from fisheries and aquaculture reached a record high of 223.2 million metric tonnes in 2022. Of this, an estimated 89%, equivalent to 20.7 kg per capita was used for human consumption [16] (Figure 1a). Furthermore, over 230 countries participated in international seafood trading, generating a record value of USD 195 billion. This marked a 19% increase compared to values before the global pandemic in 2020 [16]. In 2018, global seafood consumption was approximately 20.5 kg per capita, with aquaculture producing 82 million metric tonnes of seafood, valued at USD 250 billion [18]. By 2022, aquaculture surpassed capture fisheries, with global aquaculture production reaching 130.9 million tonnes, valued at USD 312.8 billion. Furthermore, the global apparent consumption of aquatic products in 2021 was 162.5 million tonnes, resulting in 20.6 kg per capita consumption [16]. China has maintained its position as a global leader in seafood production, consumption, imports, and exports. In 2018, the country exported around 3.46 million tonnes of seafood. By 2022, China contributed 36% to the total production of aquatic animals and accounted for 36% of the global seafood consumption, highlighting its significant influence in the worldwide seafood industry [16,19]. In 2021, Asia recorded the highest apparent consumption of aquatic animal foods at 116.1 million tonnes (71% of global consumption), aligning with its leading role in production (Figure 1b). The USA is one of the largest fishing nations due to its extensive coastlines [20] but imports over 90% of its seafood, with whitefish, salmon, shrimp, tuna, crab, clam, and oysters among the major seafood products [18,21]. Seafood consumption in the USA was estimated at 19 pounds per capita in 2019 [22]. An assessment of the retail sales of seafood from 2017 to 2019 in the USA revealed that 813,700 metric tonnes of seafood, worth $12.2 billion, were purchased from retail stores, with fresh seafood accounting for 43.3% of the revenue generated [23]. The FAO projects that seafood production will grow by 10% to 205 million tonnes, and its consumption is expected to rise by 12% to 21.3 kg per capita by 2032 [16]. This underscores the need for a deeper understanding of the benefits and harms associated with seafood consumption.

## 3. Benefits of Seafood Consumption

### 3.1. Health Benefits

Malnutrition, ranging from micronutrient deficiencies and an inadequate intake of long-chain omega-3 fatty acids to the overconsumption of energy and red and processed meats, which contribute to the rise in chronic disease, is a global problem. Transitioning from terrestrial animal-derived food products to a higher consumption of aquatic animal-sourced foods would ameliorate those issues based on their decreased calorie density, high nutrient density, and availability of the omega-3 fatty acids that lower the risk of heart disease and certain types of cancer while promoting retinal and brain health. Aquatic food production has the potential to increase with appropriate international focus and investments [24]. It is suggested that an increase in aquatic food consumption would decrease the consumption of terrestrial animal foods high in saturated fat, benefiting health; providing protein and micronutrients, reducing micronutrient deficiencies including vitamin A, calcium, iron, and zinc; and reducing the greenhouse gas emissions that are typically higher for the production of terrestrial animal foods [24]. Economically, incentivizing aquatic infrastructure and production would lower the market price and increase availability.

### 3.2. Nutritional Benefits

#### 3.2.1. Protein

Seafood is a high-protein and low-fat food of high biological value. It provides all the essential amino acids in adequate quantities. Additionally, it is more digestible than meat because of its lower connective tissue content, which is apparent in the flakiness of cooked fish [25]. According to the U.S. Department of Agriculture National Agricultural Food Data Central, the protein content ranges from approximately 10 to just under 23% [26].

#### 3.2.2. Omega-3 Fatty Acids

Not all fish have a high content of fat, but those that do, fatty fish, contain eicosatetraenoic acid (EPA, C20:5n-3) and docosahexaenoic acid (DHA, C22:6n-3). EPA and DHA are found in increasing amounts (per 100 g serving) in seafood sources such as white and light tuna (0.24–0.27 g), rainbow trout (0.84 g), sardines (1.19 g), pacific oysters (1.56 g), pacific herring (1.7 g), and various types of salmon (1–1.83 g) [26,27]. The very long omega-3 fatty acids found in cold-water fish have numerous health benefits. Although the body can synthesize EPA and DHA by desaturation and elongation of alpha-linolenic acid (ALA, C18:3n-3), this hepatic conversion is inefficient, with a rate typically less than 15% [28]. Potentially due to the effects of estrogen, the conversion rate was higher in young women with a 21% conversion rate to EPA and 9% to DHA compared to young men with a conversion rate of 8% [29], making these very long n-3 fatty acids conditionally essential.

Microalgae are the original producers of these very long fatty acids that are then concentrated in fatty fish. Through the marine food chain, they are consumed by brine shrimp, copepods, rotifers, the early larvae of crustaceans and fish, and the early and late larvae of mollusks. Some of these serve as food sources for larger seafood species [30].

#### 3.2.3. Health Benefits of Omega-3 Fatty Acids

The health benefits of these omega-3 fatty acids are wide-ranging, including anti-inflammatory effects and reduced risks of hypertension, cardiovascular and neurological diseases, and impaired brain function, such as depression. Additionally, they have immune-modulating benefits that are employed in the treatment of autoimmune diseases including rheumatoid arthritis, inflammatory bowel diseases, psoriasis, cystic fibrosis, asthma, and lupus [30]. If the brain is deficient in DHA, it increases the risk of Alzheimer’s disease, Parkinson’s disease, schizophrenia, bipolar disorder, and depression. As people age, a lack of DHA leads to a cognitive decline [31]. According to the Dietary Guidelines for Americans, 2020–2025, seafood consumption is important for pregnant women because of its cognitive benefit to the offspring. The U.S. Dietary Guidelines recommend the consumption of seafood varieties “that are higher in EPA and DHA and lower in methylmercury, which include salmon, anchovies, sardines, Pacific oysters, and trout. Tilapia, shrimp, catfish, crab, and flounder are commonly consumed varieties that also are lower in methylmercury” [3]. Those seafood sources lower in methylmercury also contain fewer n-3 fatty acids [26].

#### 3.2.4. Vitamin D and Fish

Vitamin D is a steroid hormone that is involved in the regulation of calcium and phosphorous in the human body. Its deficiency is a public health concern in the United States, although the risk of an inadequate vitamin status has decreased in recent years [32]. According to the latest NHANES study on the vitamin D status of Americans, based on data from 2001 to 2014, the incidence is highest in non-Hispanic blacks, women, those aged 20–29, and during the winter season. Fatty fish is a good source of vitamin D [33] and the Recommended Dietary Allowance (RDA) for vitamin D is 15 mcg or 600 IU for ages 14 to 70 [34].

It is difficult to consume adequate amounts of vitamin D from food. One of the few excellent sources of vitamin D includes selected fatty fish. The amount of vitamin D does not only vary significantly between different fish species but also within species caught in different areas of the globe. Importantly, this variation in the vitamin D3 content of zooplankton contributes to the vitamin D3 levels in the fish that consume them. Contrary to popular belief, the content of vitamin D is not related to the fat content in fish [35]. The vitamin D content of wild-caught salmon was significantly higher than that of farmed salmon, supporting the role of the fish’s diet on vitamin D content. Additionally, the authors reported that the amount of vitamin D published for some fish was unreliable and may require revision [36]. These findings were further confirmed, stating that the vitamin D content of wild salmon varies by region but was higher than that of farmed salmon [37]. Furthermore, the authors reported that feeding farmed salmon with vitamin D-containing feed increased the vitamin D in the fish significantly. The Current Dietary Guidelines report the following for 3 oz fish servings: 645 IU for three freshwater rainbow trout, 383–570 IU for various types of salmon, 231 IU for light canned tuna, and 182 IU for herring [3]. Low-fat fish sources such as tilapia and flounder ranged in the low one hundreds for vitamin D content [3].

#### 3.2.5. Choline

Only recently recognized as an essential nutrient in 1998 by the Institute of Medicine [38], choline is a vitamin that can be made endogenously in the human liver, but typically not in adequate quantities [39]. Existing as fat- or water-soluble biomolecules in food [40], it is a major methyl donor and a component of several phospholipids and the neurotransmitter acetylcholine. Most Americans consume approximately 150 mg less than the Adequate Intakes recommendations for choline by the Food and Nutrition Board of the Institute of Medicine, which are 425 mg/d and 550 mg/d for women and men, respectively [41]. These recommendations are based on the prevention of liver damage in choline depletion studies [40]. It is concerning that only about 10% of the U.S. population consumes adequate choline levels, including women in their reproductive years, especially considering that choline decreases the risk of birth defects including neural tube defects and cleft palate [40]. Unfortunately, prenatal supplements typically do not contain choline. Other than eggs and liver, beef and fish are the best sources of choline [39,40].

#### 3.2.6. Zinc

Zinc, a divalent cation, is a cofactor for multiple enzymes in the human body and plays a significant role in protein and DNA synthesis, the immune system, wound repair, and gene transcription [42]. Since the human body cannot store significant amounts of zinc, it needs to be consumed regularly [43]. The recommended dietary intake for adult women and men is 8 mg and 11 mg, respectively. The best dietary sources for zinc are beef and shellfish products, of which oysters have the highest content. Although plant products also contain zinc, it is less bioavailable due to the content of the phytates or oxalates that bind zinc and interfere with its absorption [42,43].

#### 3.2.7. Iodine

Iodine is an essential trace element that regulates the basal metabolic rate and plays a significant role in brain development. In many parts of the world, an adequate iodine consumption to meet the RDA of 150 mcg/d in adults remains challenging with an estimated 300 million people with mental deficits attributable to inadequate intake [44]. In the United States, many pregnant women, requiring 220 mcg/d of iodine, have an inadequate iodine intake, especially women who do not consume enough dairy, are vegetarians, or are on a salt-restricted diet [45]. Since coastal areas furnish the highest amount of iodine, seafood remains one of the best dietary sources. Seaweed, including kelp, nori, kombu, and wakame, provides the largest amount of iodine, but seafood and eggs are also very good sources. Although dairy products can be good sources, they are less reliable due to the dependence on the cow’s feed [45].

#### 3.2.8. Selenium

Selenium, an essential micronutrient, plays a significant role as an antioxidant and anti-infectious agent. The human body requires selenium to metabolize the thyroid hormones and DNA synthesis. Although selenium deficiency is rare in the United States, its availability is problematic in areas where plants are grown on selenium-poor soils and seafood is not readily available. Additionally, hemodialysis patients and HIV-positive patients frequently have low plasma selenium levels, although more research is needed to determine if this is clinically significant and can be ameliorated by an increase in consumption [46]. Fish is an excellent source of selenium, a component of 25 selenoproteins in the human body. It contains mainly the organic form of selenium, which has a higher bioavailability [47]. According to the Food and Nutrition Board, more than 90% of the selenomethionine and a high amount of the selenocysteine are absorbed by the human body. Although selenate, the inorganic form, is also almost completely absorbed, the significant urinary loss decreases its benefit to the body [48]. The bioavailability of selenium (Se) from seafood is limited by the presence of methylmercury (MeHg), which binds strongly to Se, forming MeHg–SeCys complexes that reduce Se absorption and availability [49]. There is some evidence that adequate selenium consumption may be protective against the neurotoxicity of methylmercury as selenium forms an insoluble complex with mercury, thereby reducing the danger of mercury exposure by decreasing mercury absorption [50,51].

Selenium consumption presents an interesting dilemma. While it is an essential trace mineral, it can easily become toxic since the window between adequacy and toxicity is narrow. Environmental contamination through mining and industrial waste can contaminate seafood. The typical protective “buffers” used to establish “safe” consumption levels for other toxicants, such as setting regulatory limits two orders of magnitude below the “no observed adverse effect level”, may not be applicable for selenium [50]. Since selenium is also part of many nutrient supplements, further raising intake levels, advisories on selenium limits in the United States and Canada must consider supplemental dietary selenium consumption. The overconsumption of selenium, especially when people consume large amounts of seafood or subsist on fish harvest, can lead to toxic levels, resulting in neurotoxicity and hair and nail brittleness or loss [46,50].

## 4. Harms of Seafood Consumption

### 4.1. Physical Contaminants

#### 4.1.1. Micro- and Nanoplastic Contamination of Fish and Shellfish

According to the Oxford Languages dictionary, microplastics are “extremely small pieces of plastic debris in the environment resulting from the disposal and breakdown of consumer products and industrial waste” [52]. These include the microbeads in personal care products such as polyethylene plastics in skin exfoliants and toothpastes, which have been banned in the United States. Microplastics are defined as having particles smaller than 5 mm [53]. Then microplastic contamination of the environment creates challenges for the seafood industry. The buildup of microplastics causes biological harm to the aquatic organisms themselves and introduces additional microplastics into the human diet that can have negative consequences for human health. Nanoplastics are either defined as 1–100 or 1–1000 nanometers in size [54]. Micro-nanoplastics (MNPs) are derived from plastic debris fragmentation, which is a process that all plastics sustain, through physical abrasion, heat, UV radiation from sunlight, biological processes, and/or chemical stress [55,56]. The physical properties and chemical composition of the polymer determine the level of toxicity of the MNAs [5]. The degradation of microplastics is difficult due to their hydrophobic nature and functional groups that are resistant to degradation. Degradation methods include photodegradation, thermal degradation, chemical degradation, and biodegradation [56].

##### The Effect of Microplastics on Fish and Shellfish

The United Nations GESAMP (Joint Group of Experts on the Scientific Aspects of Marine Environmental Protection) identified plastic waste as a “major stressor” of the marine environments that must be monitored to inform policy decisions. Workgroup 43 was tasked to specifically identify the plastic contamination generated by the shipping industries to reduce contamination through “LC/LP waste streams, hull paints/coatings, and abandoned fiber-reinforced polymer vessels as specific sources of plastic litter and micro-plastics in the ocean” [57]. Microplastics in rivers, lakes, and oceans, in the form of the fragments, fibers, and filaments that are most common in natural habitats, are ingested by fish and shellfish. They can cause biological harm through their physical properties, including size, shape, and particle type, and chemical harms due to their chemical properties, including their chemical composition and absorption of water contaminants such as the heavy metals responsible for toxicological effects [5,55]. In addition to their effect on the seafood themselves, they also enter the food chain and can go through a process of potential bioaccumulation. This bioaccumulation and biomagnification results from contaminated phytoplankton, zooplankton, etc., serving as food sources for seafood [55]. The effect of the accumulation of micro- and nanoplastics on marine organisms lowers steroid hormones, interferes with fertility, decreases immunity, and increases oxidative stress, thereby threatening marine life. The ingestion of the microplastics themselves and the environmental contaminants concentrated in them harm the marine ecosystem [58]. Laboratory studies on the toxicity of microplastics provide diverging results compared to the data derived from field studies, because the microplastics used in laboratory studies are more often smooth microbeads, while those in aquatic environments have irregular shapes and represent 85% of ingested microplastics. Studies on the effect of nanoplastics are sparse compared to the larger-sized microplastics [5].

##### The Effect of Microplastics on the Human Body

Micro-nanoplastics (MNPs) can enter the human body through inhalation, the consumption of contaminated water or food, or dermal contact [59]. Mollusks, as filter feeders, actively accumulate microplastics from their surrounding aquatic environments, leading to their retention in tissues. Consequently, the human consumption of these seafood species may serve as a pathway for microplastic ingestion (Figure 2). Micro- and nanoplastics have been identified in the blood, urine, feces, placenta, and breast milk [56]. Many plastics containing endocrine-disrupting chemicals are responsible for diseases since they affect the hypothalamus; pituitary, thyroid, and adrenal glands; testes; and ovaries. The toxicity of microplastics is mainly attributable to oxidative stress [56]. Moreover, microplastics’ disruption of the T3 and T4 thyroid hormone production and metabolism was identified [60]. A study of plastic packaging from five countries, including the United States, reported that “chemicals activating the pregnane X receptor (PXR), peroxisome proliferator-activated receptor γ (PPARγ), and estrogen receptor α (ERα), as well as those inhibiting the androgen receptor (AR), are prevalent in these products” [61]. According to the National Institute for Environmental Health Sciences, exposure to various endocrine disrupters occurs through the air, diet, skin, and water and, depending on the product, contributes to the incidence of ADHA and diminished immunity in children, increasing the risk of metabolic disorders including diabetes, premature breast development in puberty, and reproduction [46]. Nanoplastics, less than 1 micrometer in size, are able to cross the cellular and blood–brain barriers and accumulate in such organs as the brain, liver, and gonads [62]. Within the cell, the MNPs can damage lysosomes and cause mitochondrial dysfunction, oxidative stress, and inflammation, injuring the cell and potentially resulting in cell apoptosis [56]. The first microplastics were identified in the human placenta using Ragan microspectroscopy [63]. In addition, they have been identified in breast milk and meconium [64]. After that, several in vitro and rodent studies were reviewed, suggesting that MNPs “induce inflammation and genotoxicity, the latter being recognized as a strong predictor of carcinogenicity” [65]. Inflammation is believed to arise from the immune system’s inability to eliminate microplastics, increasing the risk of cancer and potentially contributing to the growing incidence of neurodegenerative disorders, particularly in individuals exposed to high concentrations or those with a heightened susceptibility to adverse reactions [59].

##### Microplastics as Transmitters of Heavy Metals and Microbial Pathogens

Micro- and nanoplastics are a transport medium for numerous harmful chemicals and heavy metals that serve as additives in the production of plastics [67]. Due to their large surface area and small size, microplastics adsorb toxic environmental substances, including persistent organic pollutants (polycyclic aromatic hydrocarbons, polychlorinated biphenyls, dioxins, furans, organochlorine pesticides, and polybrominated diphenyl ethers), heavy/trace metals, antibiotics, microorganisms, and others that are toxic to environmental organisms and humans at the end of the food chain [55,56].

### 4.2. Chemical Contaminants

#### 4.2.1. Heavy Metals in Fish and Shellfish

When considering the consumption of fish and shellfish, the question of heavy metals, metals with a high atomic weight and a density above 5 g/cm^3^, arises [68,69]. Some heavy metals such as iron, zinc, copper, cobalt, and manganese are essential trace elements that humans must consume in small amounts; however, they are toxic to the body at higher concentrations. Others, such as cadmium, lead, and mercury, and metalloids, such as arsenic, have no biological or physiological function [68] and are harmful to the environment and living organisms including the human body [69]. While regulatory measures control the content of these elements in food, their bioaccumulation at trace concentrations remains a concern due to potential chronic effects on human health [68].

Heavy metal content depends on an organism’s ability to concentrate them in their tissues. Large predatory fish concentrate higher amounts of those heavy metals [27]. Heavy metals enter phytoplankton and zooplankton, which transport them through rivers, lakes, and other marine environments. Through the marine food chain, they are finally exposed to the consumer [70]. The degree of heavy metal bioaccumulation depends on several factors including the water temperature; pH; dissolved oxygen; chemical form of the metal; and the fish’s habitat, type, age, gender, size, and physiology [71]. Predatory fish at the top of the food chain, such as swordfish, tilefish, king mackerel, some forms of tuna, and shark, tend to accumulate heavy metals such as mercury. These fish are either exposed to mercury, cadmium, or lead through the water or the consumption of other fish, hereby becoming increasingly contaminated with age [72,73]. In the human body, they can attach themselves to proteins by displacing the metals to which they typically bind, resulting in cellular malfunctioning, tissue damage, and systemic toxicity. Ultimately, this can cause multiple organ damage [68]. The toxic effects of heavy metals include impaired renal function (Pb, Cd, and Hg), liver damage (Pb and Cd), reduced cognitive function (Pb and Hg), impaired reproductive capacity (Cd and Pb), hypertension (Cd), neurological changes (Hg and Pb), teratogenic effects (Hg), and cancer (Cd).

##### Mercury

Mercury (Hg), predominantly in the organic form of methylmercury (MeHg), is contained in small amounts in all fish and shellfish, where it is distributed throughout the muscle tissue of the animals [28]. Mercury mainly originates from off-gassing from the earth’s crust and environmental pollution [74]. The health effects of mercury result from the interference of protein synthesis, microtubule disruption, and an increase in intracellular calcium, which disturbs neurotransmitter functioning [75], and include damage to the central nervous system, kidneys, and gastrointestinal tract [74,75]. Initial symptoms range from paresthesia to blurred vision and malaise [72,76]. Infants exposed to high mercury levels during pregnancy display mental retardation, ataxia, visual field constriction/blindness, deafness, and cerebral palsy. Lower levels of exposure still lead to developmental delays and abnormal reflexes [76]. Mercury uptake by fish primarily occurs through dietary exposure, accounting for about 80–90% of their total mercury intake. The bioavailability of Hg in fish varies significantly depending on its chemical form, with MeHg being assimilated more efficiently (10–100%) than Hg (II) (2–51%) [77]. In humans, mercury exposure mainly arises from consuming fish contaminated with MeHg, which is typically more bioaccessible (2–100%) compared to Hg (II) (0.2–94%). Estimates from several studies showed that the actual systemic absorption rates are consistently lower (12–79% for MeHg and 49–69% for Hg(II)), indicating that previous bioaccessibility assumptions often overestimated the actual uptake into the systemic circulation [77]. Due to its lipophilic properties, mercury can cross both the placenta and the blood–brain barrier, meaning that it is harmful at any stage of the life cycle [72,75], but especially harmful to the fetus, infants, and young children [28]. Since the concentration of mercury in fish depends on bioaccumulation and biomagnification, the same fish species may be safe if harvested in one location and not if obtained from another one [75]. The Joint FAO/WHO Expert Committee on Food Additives suggests the general population limit their methyl mercury intake to the “provisional tolerable weekly intake” of 1.6 mcg/kg body weight/week [72,76].

##### Lead

Lead poisoning is infrequently caused by fish and shellfish consumption. According to the EPA, “lead does not appear to bioconcentrate significantly in fish but does in some shellfish such as mussels”. However, since lead in shellfish concentrates in the mucus on the epidermis, dermis, and scales, eating those shellfish appears to be benign [78]. Previous studies confirmed that the amount of lead in Italian fish and seafood products was highest in squid and blue mussels but remained well below the provisional tolerable intake level of 3 mg of lead per person [79]. An evaluation of lead levels in several seafood samples, including bivalve mollusks, crustaceans, and echinoderms, from around the world found that the mean lead concentrations ranged from 0.02 to 0.39 mg/kg, with almost all samples below the maximum limit set by the European Food Safety Authority (EFSA) [80]. Another study also reported the highest bivalve mollusk lead levels in sea urchins (0.203 mg/kg), followed by scallops (0.191 mg/kg) and mussels (0.174 mg/kg) in Central Italy [81]. In 2011, the WHO withdrew its provisional tolerable intake level because they were unable to establish an amount “that would be considered health-protective” [82].

##### Cadmium

Cadmium exposure can occur through the consumption of shellfish, mussels, and dried seaweed, although cigarette smoke is the main culprit of cadmium exposure to the human body [68,83]. The FDA sets the interim reference level for cadmium at 0.21–0.36 mcg/kg bw/d, which is the level above which food may be considered harmful to human health [84]. The European guidance for a tolerable weekly cadmium intake is 0.35 mcg/kg bw/d [85]. The average oyster cadmium level is 2.2 ppm wet weight, resulting in Health Canada recommendations of no more than 12 oysters per month for adults and 1.5 oysters per month for children [86]. An Italian study confirmed that the cadmium levels in oysters, gastropods, and scallops are often higher with middle bound averages of 0.218 mg/kg, 0.217 mg/kg, and 0.117 mg/kg, respectively [81]. The concentration of cadmium in six aquatic food groups in Zhejiang, China, showed the highest cadmium concentration in marine crustaceans (1.47 mg/kg) followed by mollusks (0.925 mg/kg), confirming that aquatic food, specifically shellfish, is one of the main sources of dietary cadmium exposure [87]. A Long Island study on seafood consumption and blood cadmium levels of avid seafood consumers reported that only salmon intake in cups per week marginally increased blood cadmium levels [88]. Chronic low-level exposure to cadmium can lead to a decrease in bone density and osteoporosis by interfering with the osteoblast activities while stimulating osteoclasts [47]. When cadmium builds up in the kidneys, it causes renal disease [83].

#### 4.2.2. Per- and Polyfluoroalkyl Substances

Per- and polyfluoroalkyl substances (PFASs) are a group of man-made chemicals that have been manufactured since the 1940s for various products and industrial uses. These substances are characterized by a hydrophobic -CF_2_ chain and a hydrophilic functional head, which contribute to their exceptional chemical and thermal stability, a result of strong carbon–fluorine covalent bonds. Emissions from the production, use, and disposal of products containing PFASs have led to their extensive presence in the environment [89]. They exhibit a higher persistence in the environment than polychlorinated biphenyls (PCBs), leading to the long-term contamination of both aquatic and terrestrial ecosystems. PFASs can infiltrate drinking water and accumulate in seafood, contributing to dietary exposure. According to a 2020 scientific opinion by the EFSA, seafood was identified as the primary dietary source of exposure to perfluorooctane sulfonic acid (PFOS), which accumulates through marine and freshwater food chains, as well as perfluorooctanoic acid (PFOA) [90]. PFASs are now considered widespread contaminants found in both urban and natural settings and can enter the human body through various exposure routes [89].

Although the PFAS levels in freshwater fish generally surpass those found in commercially available fish, there has been a 30% reduction in PFOS levels (used as an indicator) between 2008–2009 and 2013–2014 [91]. The U.S. Environmental Protection Agency’s (EPA) 2018–2019 National Rivers and Streams Assessment also reported an additional 6.7% decline in PFOS concentrations in fish fillet composite samples from 2013 to 2014 [92]. Since these findings were newly released as of December 2023, further examination of the data will be necessary to gain a more comprehensive understanding of the current PFAS exposure levels in freshwater fish [93].

The U.S. Environmental Protection Agency (EPA) and the Food and Drug Administration (FDA) have not provided specific guidance on fish consumption related to PFAS exposure, leaving state and Tribal health agencies to fill this critical gap. These agencies and advisory organizations play an essential role in appropriately assessing risk and educating the public on the benefits and potential risks of fish consumption. The value of fish as a source of nutrition, as well as its cultural and food security significance, is well documented and holds particular importance for traditional and subsistence communities [94].

In 2016, the EPA set reference doses (RfDs) for perfluorooctane sulfonic acid (PFOS) and perfluorooctanoic acid (PFOA) at 20 ng/kg/day, based on the developmental impacts observed in animal studies. In April 2024, the EPA revised these RfDs for several PFASs, including lowering the RfDs for PFOS to 0.1 ng/kg/day and for PFOA to 0.03 ng/kg/day, representing significant reductions. The RfDs indicate the estimated level of daily oral exposure over a lifetime that is unlikely to result in adverse non-cancer health effects, even for vulnerable populations [94]. To establish these RfDs, the EPA prioritized epidemiological studies that linked PFAS exposure with outcomes such as reduced birth weight, diminished vaccine antibody responses in children, and increased liver enzyme (ALT) and total cholesterol levels. The EPA expressed high confidence in the developmental studies and moderate confidence in studies focused on other health outcomes, selecting those with the strongest evidence and minimal bias [94].

##### Adverse Health Effects of Per- and Polyfluoroalkyl Substances

Human exposure to PFASs is a significant concern due to their toxic health impacts, which include immune system suppression, thyroid disease, pregnancy-induced hypertension, and certain cancers [95,96]. The most harmful effects are associated with long-chain perfluoroalkyl acids (PFAAs) with six or more carbon atoms, such as perfluorooctane sulfonic acid (PFOS) and perfluorooctanoic acid (PFOA). These substances were voluntarily phased out by industry through the USEPA Stewardship Program [97,98,99]. Despite stricter regulations on PFASs globally, human biomonitoring studies continue to reveal widespread exposure to legacy PFAS compounds [96]. Furthermore, short-chain PFASs and other emerging replacement compounds are increasingly being detected [100,101].

##### Seafoods as a Source of PFAS Compounds

Seafood, including fish and shellfish, is frequently identified as a major source of non-occupational PFAS exposure for humans [95,96,102,103,104]. Studies have reported PFAS concentrations in seafood available within the U.S. For example, a recent study in Washington, D.C., analyzed 81 seafood samples from retail stores for 20 PFAS types and found the highest total concentration (23 ng/g) in canned clams from Asia, where PFOA was the dominant compound [105]. Another study assessed 26 PFAS compounds in 70 seafood samples from grocery stores and found total PFAS concentrations ranging from 0.50 to 22 ng/g, with the highest levels in walleye (*Sander vitreus*) sourced from Lake Erie [106]. Moreover, 11 PFAS compounds were investigated in 39 fish samples collected from three river sites in South Carolina, finding total concentrations between 6.2 and 24 ng/g, with the highest levels detected in spot (*Leiostomus xanthurus*), a fish commonly consumed by the Gullah Geechee community and other regional fishers [107]. Overall, these studies show that PFOS, PFOA, and PFUnDA are frequently detected at higher concentrations, while other PFAS compounds are often present at low or undetectable levels. However, current data mainly focus on a limited range of PFASs, particularly PFAAs and their precursors, with few studies addressing other emerging chemicals of concern. Moreover, most research has explored PFAS levels in seafood without considering how different dietary choices may affect consumer exposure [99].

#### 4.2.3. Biogen Amines in Seafoods

Biogenic amines (BAs) are organic compounds characterized by their low molecular weight [108,109]. In seafood products, their formation is primarily facilitated by bacterial histidine decarboxylase (HDC), an enzyme that catalyzes the decarboxylation of free histidine, contributing significantly to seafood spoilage [110,111,112]. The presence of BAs in seafood poses a major public health risk due to their physiological and toxicological effects [113]. Common BAs identified in various food products include histamine, tyramine, phenylethylamine, putrescine, agmatine, cadaverine, spermine, spermidine, and tryptamine. Among these, histamine is frequently reported in a range of food items, including fish, fish products, fermented meats, vegetables, dairy products, and alcoholic beverages [111,113,114].

Histamine, with a molecular formula of C_5_H_9_N_3_ and a molecular weight of 111.14 g/mol [110,115], is a prominent agent in scombroid poisoning and is classified as a chemical hazard in food safety. The accumulation of histamine in seafood occurs when histamine-producing bacteria proliferate under high ambient temperatures [116]. These bacteria produce HDC, which converts free histidine to histamine [116]. Post-harvest handling and improper storage conditions frequently contribute to contamination by histamine-producing bacteria. High histamine levels are commonly found in fish species from the Scombroidae family [117,118,119,120], thus leading to the term “scombroid poisoning” when referring to the foodborne illness caused by these species [120]. However, non-scombroid fish species, such as mahi-mahi, anchovy, amberjack, marlin, bluefish, herring, and sardine, have also been implicated in scombroid poisoning. Fish with darker muscle tissues, including tuna, blue scad, chub mackerel, bonito, and saury, may also accumulate significant amounts of histamine due to their high levels of free histidine [110,111,114,121,122,123,124,125,126,127]. International regulations regarding the maximum allowable histamine levels in fish and fishery products vary, with the specific limits defined by the sample size (n), the permitted number of defective units within the sample (c), and acceptance limits (m). Previous studies have recommended that histamine concentrations up to 50 ppm are normal and safe for human consumption, while levels between 50 and 200 ppm indicate mishandling and potential toxicity. Concentrations ranging from 200 to 1000 ppm are classified as unsatisfactory and likely toxic, and any levels exceeding 1000 ppm are definitively toxic and unsafe [128]. In fresh or fresh-frozen fish, the maximum allowable histamine levels range from 30 ppm (n = 18, c = 0, m = 30 ppm), as recommended by the National Fisheries Institute (NFI) for U.S. tuna canneries, up to 400 ppm in China. The EU, Australia, China, Egypt, Mexico, New Zealand, South Korea, Taiwan, and Turkey have established a maximum histamine limit of 200 ppm for canned fish products (EU criteria: n = 9, c = 2, with no more than two samples exceeding 100 ppm, and none exceeding 200 ppm) [128].

##### Effects of Biogen Amines on Human Health

The clinical symptoms of histamine poisoning typically manifest within minutes to a few hours post-consumption of contaminated fish and involve various organ systems, leading to both gastrointestinal and systemic symptoms [127,129,130]. These range from mild reactions, such as rash, urticaria, nausea, vomiting, diarrhea, flushing, and headache, to tingling and itching of the skin. In cases where foods containing elevated histamine levels are consumed, the enzymes in the intestines responsible for histamine degradation may be insufficient, resulting in histamine absorption into the bloodstream and the onset of neurological, gastrointestinal, and respiratory symptoms [110,111,127,129]. The duration of these symptoms can vary, often lasting from several hours to days, depending on the histamine dose and individual sensitivity [111,115,127].

Histamine also acts as a multi-functional mediator in disorders such as psoriasis, where increased histamine levels are implicated [131]. In the human body, mast cells and basophils—key components of the immune system—are the primary sources of histamine, which they release rapidly in response to immune triggers [132]. These cells store high concentrations of histamine, which can be utilized in diagnostic assessments for allergies (e.g., histamine release tests) [133]. Histamine production and release by various immune cells are mediated by the HDC enzyme’s decarboxylation of free histidine [134].

#### 4.2.4. Allergens in Seafoods

Seafood allergies are a significant public health concern, affecting millions of individuals worldwide. Fish allergies are primarily triggered by parvalbumins, a type of calcium-binding protein that is highly heat-stable and resistant to digestion, making it a potent allergen capable of causing severe reactions even after cooking. In addition to parvalbumins, other allergens that can induce fish allergies include enolases and aldolases, which are enzymes involved in cellular metabolism, and fish gelatin, derived from fish collagen, which is often used in processed foods and pharmaceuticals. Shellfish allergies, which include allergies to both crustaceans and mollusks, are primarily caused by tropomyosins, muscle proteins that play a critical role in muscle contraction. These proteins are also heat-stable, allowing them to retain their allergenic properties even after cooking. Other significant allergens found in shellfish include myosin light chain, another muscle protein that contributes to allergic responses; arginine kinase, an enzyme involved in energy transfer; and hemocyanin, a copper-containing protein in mollusks responsible for oxygen transport. These allergens can elicit strong immune responses, potentially leading to life-threatening reactions such as anaphylaxis. Given the prevalence and severity of seafood allergies, understanding the nature of these allergens is essential for improving allergen management, developing accurate diagnostic methods, and enhancing food safety practices [135,136,137].

Fish and shellfish are known to trigger significant immunoglobulin E (IgE)-mediated allergic responses in susceptible individuals. These allergic reactions are associated with three primary phyla: Chordata (finned fish), Mollusca (including molluscan shellfish and cephalopods such as mussels and squids), and Arthropoda (crustacean shellfish, including shrimp and crab) (Table 1). The primary allergens responsible for seafood allergies are high molecular weight proteins that are resistant to heat and remain active even after cooking. The most common clinical symptoms of a seafood allergy include oral allergy syndrome, urticaria/angioedema, gastrointestinal disturbances, and, in severe cases, life-threatening anaphylaxis [135]. Reactions typically occur within two hours of exposure, although delayed responses up to eight hours have been documented, particularly among individuals with shellfish allergies [136,137].

Notably, approximately 11% of fish-allergic individuals in Spain and 30% in South Africa reported repeated allergic reactions triggered by the accidental inhalation of seafood vapors, despite adhering to strict avoidance diets [138,139]. Seafood allergy is also a leading cause of food-induced anaphylaxis in the United States, Canada [140,141], and parts of Europe [142] and is particularly prevalent in Southeast Asia [143] and Australia [144]. Approximately 20% of Australian children with seafood allergies have reported a history of anaphylaxis [144]. Additionally, exercise-induced anaphylaxis has been observed following the consumption of certain shellfish, such as shrimp, squid, and oysters [145,146]. These findings underscore the critical importance of the accurate diagnosis and effective management of seafood allergies, especially considering that seafood allergies are generally lifelong [139].

**Table 1 foods-14-01461-t001:** Seafoods classification causing allergic symptoms (Adopted from [147]).

Phylum	Class	Common Name
Shellfish Mollusca	Gastropoda Bivalvia	Abalone Mussels, scallops, clams, and oysters
Arthropoda	Cephalopoda Crustacea	Squids, cuttlefish, octopus, lobster, crayfish, shrimp, prawn, and crab
Fish Chordata	Bony fish-Osteichthyes	Tuna, hake, cod, herring, salmon, sole, pilchard, anchovy, yellowfin, trout, and swordfish

##### Effects of Allergens on Human Health

Patients diagnosed with fish or shellfish allergies are generally advised to completely avoid consuming the respective type of seafood. However, beyond true IgE-mediated seafood allergies, non-IgE-mediated reactions, such as histamine fish poisoning (HFP) and histamine intolerance, can produce allergy-like symptoms. The management of these conditions does not typically require the complete dietary avoidance of seafood. Given that seafood is a valuable source of omega-3 fatty acids, which confer multiple health benefits, optimizing seafood consumption remains essential. This manuscript aims to discuss the differential diagnosis and treatment approaches for seafood allergy and related conditions [147].

The diagnosis of seafood allergy typically relies on a thorough evaluation of the patient’s medical history, skin prick testing (SPT), the quantification of specific IgE antibodies, and, in certain cases, oral food challenge tests. SPT is commonly employed as a quick initial screening tool for seafood allergy; however, its predictive accuracy is limited. Various commercial extracts of fish and shellfish, as well as specific allergens, are available for measuring specific IgE antibodies. The double-blind, placebo-controlled food challenge remains the gold standard for diagnosing food allergies, including seafood allergy, as it provides essential guidance for patients on dietary restrictions [147,148,149]. Diagnosing seafood allergy poses significant challenges for clinicians, as a positive SPT or detectable levels of specific IgE do not conclusively indicate clinical reactivity. A diagnostic decision tree for assessing seafood allergy in patients has been proposed in other studies [150].

### 4.3. Biological Contaminants

#### 4.3.1. Foodborne Pathogens in Seafood and Their Implications on Human Health

Foodborne pathogens and their toxins are one of the major threats from the consumption of seafood [151]. The contamination of seafood with foodborne pathogens is influenced by several factors, including the microbiological quality of harvest water; environmental conditions; the proximity of harvesting areas to sewage; the feeding habits of aquatic animals; and the practices involved in harvesting, handling, processing, and preparation of seafood, as well as the time period [151,152]. For example, the data presented in Figure 3 suggest a clear shift in the source of seafood-related outbreaks over time. While fish were the predominant source of outbreaks in the earlier period (1973–1979), their contribution gradually decreased, with mollusks increasingly becoming the primary source of outbreaks in the later periods, particularly from 1990 onward. The consistent yet relatively smaller contribution from crustaceans indicates that they have played a minor but steady role in seafood-related outbreaks. This trend highlights the growing importance of monitoring and regulating mollusks to prevent outbreaks, as their share of incidents has risen significantly over the years. Additionally, the decline in fish-related outbreaks may suggest improved safety practices in the handling and processing of fish or changes in consumption patterns. Seafood-related infections can be triggered by various foodborne pathogens including viruses, bacteria, and biotoxins from harmful algal blooms [152]. These foodborne pathogens have been identified in seafood at several locations (Table 2) and have caused multiple foodborne outbreaks globally (Table 3).

##### Viruses as Pathogens in Seafood

Viruses have a smaller structure and an external protein coat protecting their genetic material. Millions of viruses are present in just a milliliter of seawater, and over 7000 viral units can exist in disinfected secondary effluents, posing a risk of severe illnesses such as gastroenteritis and hepatitis [155]. According to a recent surveillance summary by the Centers for Disease Control and Prevention (CDC), viruses accounted for 23.2% of the 405 reported foodborne illness outbreaks in the United States between 2020 and 2022 [156]. Shellfish, due to their filter-feeding nature, can harbor viruses in their guts for up to ten weeks without a significant decline in quality [157,158]. These viruses can proliferate rapidly within the shellfish, particularly when proper handling, transportation, and commercialization practices are not followed [159]. Although techniques such as depuration have been shown to reduce bacterial contaminants in these shellfish, this is not true for viruses [160,161].

Common viruses associated with seafood-borne outbreaks include norovirus (NoV), hepatitis A virus (HAV), hepatitis E virus (HEV), adenovirus (AdV), rotavirus (RV), sapovirus (SaV), and aichivirus (AiV) [159,162]. Norovirus (NoV) is the leading cause of foodborne illness outbreaks in the USA, responsible for approximately 2500 reported outbreaks annually, many of which result from consuming contaminated shellfish. The symptoms include stomach pain, nausea, diarrhea, vomiting, headaches, and fever [163]. Shellfish can accumulate NoV at concentrations 10 to 1000 times higher than their surrounding environment, and it is estimated to be responsible for 9–34% of foodborne noroviruses cases [56,164]. HAV can persist in low-temperature environments, such as cold storage for seafood, maintaining its infectivity [165]. They cause human acute viral hepatitis with symptoms including nausea, anorexia, abdominal pains, and malaise [160]. Shellfish raised in HAV-contaminated waters can carry approximately 10 to 100 viral particles and were identified as the source of 46 HAV outbreaks between 1986 and 2012 [7]. Both NoV and HAV, resilient under various environmental conditions, are linked to approximately 200,000 global deaths annually, mostly from acute gastroenteritis [166]. HEV has emerged as a significant concern in industrialized regions, where it is commonly transmitted through contaminated foods. Approximately 20 million people worldwide are newly infected with HEV each year, with an estimated 3.3 million developing acute hepatitis E and about 70,000 dying [6]. Although the virus has about eight genotypes, only HEV-3 and HEV-4 are zoonotic and found in animal and sporadic human infection cases [167]. Although the primary reservoirs for the virus include undercooked or raw pork and meat from non-domesticated animals, seafood, particularly raw shellfish harvested from waters contaminated with feces from HEV-infected livestock or humans, also transmit the virus to humans [6,7].

##### Bacteria as Pathogens in Seafood and Their Antimicrobial Resistance

Bacteria are significant causes of foodborne illnesses, accounting for 48.4% of the 405 outbreaks reported between 2020 and 2022 in the USA [163]. Globally, foodborne bacterial contamination leads to over half a billion cases of illnesses annually, with about half a million resulting in fatalities. This impact is particularly severe in children under 5 and immunocompromised elderly individuals. Mortality rates from foodborne bacterial infections are heightened by antimicrobial resistance (AMR), a growing concern in recent years [168]. While bacterial infections are typically treated with antibiotics, prolonged exposure can result in the development of resistance mechanisms or intrinsic traits in bacteria that render them unresponsive to one or multiple antibiotics. In aquatic environments, AMR bacteria can be introduced through the runoff from terrestrial land, the discharge of antimicrobial compounds such as disinfectants and antibiotics, and contamination from the fecal waste of antibiotic-treated livestock [169]. Bacteria such as *Vibrio*, *Shewanella*, *Pseudomonas*, *Clostridium*, *Aeromonas*, *Photobacterium*, and *Bacillus* are common to aquatic environments, while *Enterococcus*, *coliforms, Escherichia coli*, *Salmonella*, *Shigella*, *Listeria*, *Staphylococcus*, *Campylobacter*, *Enterobacter*, and *Klebsiella* are among those that accumulate in aquatic environments and seafood through contamination from animal and human waste [170,171].

*Vibrio* is a Gram-negative, halophilic, and facultative anaerobic bacterium that naturally inhabits estuarine and coastal environments [172]. Among its species, *V. vulnificus*, *V. cholerae*, and *V. parahaemolyticus* contribute significantly to most cases of human vibriosis [173,174]. Typically, *Vibrio* infections are associated with consuming contaminated raw or undercooked seafood [175]. In the USA, *Vibrio* is responsible for an estimated 80,000 illnesses and 100 deaths annually, with *V. parahaemolyticus* accounting for the majority of these infections and *V. vulnificus* being the most lethal [175,176]. *V. cholerae*, the etiological agent of cholera, prevalent in Africa and Asia, can thrive in fresh and brackish waters [177]. There are significant concerns regarding the emergence of antibiotic-resistant *Vibrio* in seafood. It was reported that approximately 86% of *V. parahaemolyticus* isolates from retail marine fish, oysters, shrimp, and squid samples in Vietnam were resistant to at least one antibiotic [178]. The highest rate of resistance was to ampicillin (81.43%), followed by cefotaxime (11.43%), ceftazidime (11.43%), trimethoprim-sulfamethoxazole (8.57%), and tetracycline (2.86%). In the USA, it was reported that the ampicillin resistance of *Vibrio* spp. isolated from salmon, shrimp, and tilapia from retail grocery outlets across eight states in 2019 were 42.1, 43.3, and 26.7%, respectively [128]. The isolates from shrimp further showed resistance to cefoxitin (1.7%), ciprofloxacin (0.6%), gentamicin (1.1%), meropenem (1.7%), tetracycline (5.5%), and trimethoprim-sulfamethoxazole (0.6%). Additionally, a study in South Africa showed that *V. cholerae* isolates obtained from prawn, crab, and mussels sampled from fish markets, freshwater, and brackish water exhibited high resistance against polymyxin, ampicillin, and amoxicillin/clavulanate. Although *V. mimicus* isolates were found to be completely susceptible to amikacin, gentamycin, and chloramphenicol, the isolates exhibited relatively high resistance against nitrofurantoin, ampicillin, and polymyxin [177].

*Shewanella* is another indigenous marine bacterium that can adapt to various physiological and respiratory conditions and causes infections such as bacteremia, soft tissue and wound infections, arthritis, peritonitis, osteomyelitis, and discitis [179,180]. Among the 62 species of *Shewanella* identified so far, *S. putrefaciens*, *S. haliotis*, *S. xiamenensis*, and *S. algae* are primarily associated with human infections, with *S. algae* being the most virulent [181,182]. Although *Shewanella* infections may occur in individuals without direct exposure to marine environments [183], they typically result from contact with or the consumption of contaminated seawater and seafood, as observed in a case study in Japan where 69.2% of patients with *S. algae* infections had previously handled fresh fish [184]. *Shewanella* spp., the primary agents of spoilage in fish and seafood, even at cold storage temperatures [185], are responsible for diseases and massive mortalities in aquatic animals such as rainbow trout, common carp, tongue sole, red drum, Fujian oysters, and mangrove clams [186,187]. The resistance of *Shewanella* spp. to antimicrobial agents was reported in seven isolates from fresh sea urchins (*Paracentrotus lividus*) in France. Four of these isolates exhibited resistance to aztreonam, ceftazidime, and nalidixic acid, while three showed resistance to cephalexin, cefepime, amikacin, and tobramycin. Additionally, one isolate demonstrated resistance to ampicillin-sulbactam, piperacillin-tazobactam, ertapenem, imipenem, erythromycin, ofloxacin, and tetracycline [188]. Almost 42% of thirty-three isolates of *S. baltica* from wild Atlantic mackerel showed the presence of the novel mobile colistin resistance gene *mcr-4.3* [189]. In Korea, 16 *S. putrefaciens* strains isolated from the shellfish were completely susceptible to chloramphenicol, sulphamethoxazole-trimethoprim, tetracycline, ciprofloxacin, cefepime, and cefotetan but exhibited high resistance to vancomycin (87.5%), cephalothin (75.0%), ampicillin (43.8%), and streptomycin (43.8%) [190].

*Pseudomonas* is a Gram-negative bacterium that inhabits diverse environments and produces complex biofilms that provide protection and enhance its resistance against environmental stress and disinfectants [191,192]. In aquaculture and seafood processing facilities, this bacterium contributes significantly to disease and mortality in aquatic species, in addition to spoilage and the degradation of seafood quality [160,193]. In humans, it causes several infections including nosocomial infections, pneumonia, urinary tract infections, wounds, and bloodstream infections [194]. *P. aeruginosa*, the main species associated with human infections, is among the top three causes of opportunistic infections in humans, affecting over 2 million patients and causing about 90,000 mortalities annually [195]. The prevalence of antimicrobial resistance of *Pseudomonas* in seafood gives rise to concerns over its multi-drug resistance [196]. In the USA, an estimated 32,600 infections and 2700 deaths in 2017 were attributed to the multi-drug-resistant *P. aeruginosa* [197]. From a salmon processing plant in Norway, it was reported that 92 and 87% of the *Pseudomonas* isolates were resistant to ampicillin and amoxicillin, respectively [198]. Additionally, 86% of the isolates obtained from samples collected at various points, including the conveyor belt in the slaughter department, the suction area of the gutting machine, and the inlet water, exhibited multi-drug resistance to numerous antibiotics including ampicillin, amoxicillin, oxolinic acid, florfenicol, doxycycline, piperacillin/tazobactam, and ciprofloxacin. An assessment of the antimicrobial resistance of *P. aeruginosa* isolates from fresh shrimps in Iran also showed that all the isolates were resistant to at least one antibiotic, with the highest resistance (88%) being to penicillin and the lowest (5%) to nitrofurantoin [199]. The concentrations of presumptive *Pseudomonas* in 50% of the fresh fillet samples of *Salmo salar*, *Gadus morhua*, and *Pleuronectes platessa* were at least 10^4^–10^5^ cfu/g in Italy [191]. The study further reported that a resistance rate of more than 50% was observed for penicillin, ampicillin, amoxicillin, tetracycline, erythromycin, vancomycin, clindamycin, and trimethoprim, with over 76% of *P. fluorescens* strains exhibiting multi-drug resistance abilities.

As a spore former, *Clostridium botulinum* produces toxins that are heat-resistant and can survive in low temperatures. These toxins cause foodborne botulism, an intoxication characterized by fatigue, dizziness, nausea, dyspnea, and muscle weakness [200,201]. *C. botulinum* type E toxins have been found in fish such as flounder, cod, rockfish, and white fish. Botulism outbreaks have been associated with whole, processed, traditional, fermented, salt-cured, and home-canned fish. Ingesting even a small amount of these toxins can lead to severe illnesses and fatalities [202]. The highly resistant endospores produced by *C. perfringens* under unfavorable conditions are less sensitive to heat and can survive temperatures as high as 100 °C for up to two hours [203]. *C. perfringens* showed high survival rates and could be isolated from clams boiled at 100 °C for five minutes [181]. This heat-resistant characteristic enhances its ability to cause foodborne infections, contributing to nearly one million foodborne illnesses annually in the USA, and it has been associated with about 13% of gastrointestinal foodborne outbreaks [204,205]. Additionally, there have been reports of the antimicrobial resistance of *Clostridium* spp. to various antibiotics. The *C. difficile* strains isolated from mussels and clams from the North Adriatic Italian Sea were sensitive to metronidazole; however, six, including five toxigenic strains, showed multi-drug resistance patterns to antibiotics such as erythromycin, moxifloxacin, rifampicin, and clindamycin [206]. Four *C. perfringens* type A isolates were recovered with both *cpa^+^* and *cpb2^+^* genes from fish and their environmental sources in India. The resistance of these isolates to ceftriaxone (100%), ampicillin (75%), tetracycline (75%), co-trimoxazole (75%), and ceftazidime (50%) was reported. Nevertheless, all of the isolates were susceptible to norfloxacin, ciprofloxacin, and amoxicillin/clavulinic acid [207].

*Escherichia coli* is commonly found in the intestines of humans and animals and hence is used as an indicator of fecal contamination in harvest waters and seafood [208]. The bacterium, isolated in the seafood harvested from sewage-polluted waters, causes severe gastrointestinal infections in humans, with symptoms such as inflammation, stomach cramps, vomiting, diarrhea, and hemolytic uremic syndrome [208,209]. Globally, *E. coli* is a significant cause of death from gastroenteritis, predominantly among children under five [210]. Pathogenic strains of *E. coli* are categorized into six main groups: enteropathogenic *E. coli* (EPEC), enterotoxigenic *E. coli* (ETEC), enteroinvasive *E. coli* (EIEC), enteroaggregative *E. coli* (EAggEC), diffusely adherent *E. coli* (DAEC), and enterohemorrhagic *E. coli* (EHEC). EHEC includes verocytotoxin-producing or Shiga-toxin-producing *E. coli* (VTEC/STEC) [211,212]. EHEC was identified as the predominant group in fresh seafood, including finfish and shellfish, in a study in India [213]. Over 200 million serotypes of *E. coli* possess the *stx1* or more potent *stx2* genes that code for Shiga toxins [211]. In the USA, it has been estimated that Shiga-toxin-producing *E. coli* (STEC) causes approximately 265,000 illnesses and an economic loss of USD 280 million each year [214]. In 2021, thirty European countries reported 6534 confirmed cases of STEC infections, marking a 37.5% increase compared to the previous year [215]. In Europe, the assessment of *E. coli* is a significant component of the sanitary evaluation for designated shellfish production areas. For Class A production zones, the acceptable limits are ≤230 MPN/100 g of flesh and intervalvular fluid in 80% of the samples [159,216]. In 2018, the Annual Report of the European Antimicrobial Resistance Surveillance Network (EARS-Net) showed that over 50% of *E. coli* isolates in Europe were resistant to at least one class of antimicrobials, with aminopenicillin, fluoroquinolones, third-generation cephalosporins, and aminoglycosides among the most prevalent [217]. In Italy, 75% of the 79 *E. coli* isolates from manila clams, striped clams, mussels, and water samples were resistant to at least a single antimicrobial agent and 38% showed multi-drug resistance abilities [218]. Resistance was commonly found to ampicillin (56%), streptomycin (52%), sulfonamides (30%), and ceftiofur (24%). Fish, shellfish, mollusks, and crustaceans from wholesale and retail markets in Korea showed resistance to tetracycline (30.7%), streptomycin (12.8%), cephalothin (11.7%), ampicillin (6.7%), and ticarcillin (6.1%) but were sensitive to amikacin, amoxicillin/clavulanic acid, and cefoxitin [219]. Additionally, STEC isolates from shrimps, crabs, and oysters in Egypt were completely resistant to multiple antimicrobial agents including penicillins, amoxycillin/clavulanic acid, colistin, fosfomycin, ceftriaxone, ciprofloxacin, and tetracycline [220].

*Salmonella* spp. are commonly found in the intestines of animals and humans and are non-native to aquatic environments. They enter aquatic ecosystems through the improper disposal of human waste, the proximity of harvest waters to sewage effluents, agricultural runoff, and contamination from livestock and wild birds [216,221]. Globally, more than 2600 serovars of *Salmonella* spp. have been identified, nearly all of which can cause illness in humans and animals [222]. These serovars are primarily associated with two main species, *S. Bongori* and *S. Enterica*. The subspecies *enterica* of *S. Enterica* alone comprises about 1547 serovars, 99% of which are pathogenic in humans and animals [223]. The virulence mechanisms of *Salmonella* spp. enable them to evade the immune defenses of hosts and cause infections with the symptoms of abdominal cramps, diarrhea, inflammation, fever, and vomiting. In severe cases, it can progress to paratyphoid or typhoid fever, depending on the strain and severity of the infection [159,200]. Gastroenteritis is the most common *Salmonellae* infection globally, with past estimates showing about 93.8 million cases resulting in 155,000 deaths annually [224]. It causes about 1.35 million infections, 26,500 hospitalizations, and 420 mortalities each year in the USA [159]. In Europe, it is the second most common cause of human gastroenteritis. According to a 2015 EU summary, among the pathogens responsible for verified seafood outbreaks, *Salmonellae* ranked third in prevalence at 12.5%, following histamine (52.5%) and caliciviruses (25%) [216]. Recently, there has been a surge in reports of multi-drug-resistant *Salmonella* spp., presenting a significant public health concern [222]. In Poland, it was reported that, although most (74.2%) of the *Salmonella* spp. isolated from clams, mussels, oysters, and scallops were sensitive to all the antimicrobials tested, some strains exhibited resistance to colistin (12.9%), ampicillin (9.7%), tetracycline (9.7%), and sulfamethoxazole (9.7%) and three of the isolates showed multi-resistant patterns [225]. The ready-to-eat shrimps sampled from open markets in Nigeria contained *S.* Enteritidis (24.4%), *S.* Typhimurium (31.1%), and other *Salmonella* spp. (44.4%). While all the isolates were completely sensitive to cefotaxime, cephalothin, colistin, and polymyxin B, they displayed resistance to penicillin and erythromycin. Among the *S.* Enteritidis and *S.* Typhimurium isolates, 64.3 and 27.3% were found to be multi-resistant to five and eleven antibiotics, respectively [226]. Furthermore, 59.9% of *Salmonella* strains in Malaysian fish were antibiotic-resistant, with a 100% resistance rate to clindamycin and rifampicin [227].

*Shigella* can survive in aquatic environments for up to six months and withstand high acidity and extreme temperatures, enhancing its adaptability to acidic stomach environments and increasing its infectivity in humans [171,228]. It is salt-tolerant, heat-sensitive, and can survive in low humidity and temperatures ranging from as high as 60 °C for 5 min to as low as 10 °C [229,230]. It is commonly associated with ready-to-eat foods such as raw meats, vegetable salads, smoked fish, and raw oysters [231]. *Shigella* has a low infectious dose, with just 10 to 100 cells sufficient to cause infection [232]. Shigellosis, its resulting infection, spreads through waterborne and fecal–oral transmission and can lead to severe symptoms such as fever, abdominal pain, bloody diarrhea, nausea, watery stool, and tenesmus [200,228]. Globally, *Shigella* causes approximately 80–160 million infections and 600,000 deaths annually, especially in children [233]. In the USA, *Shigella* ranks as the third most common bacterial pathogen in human enteric infections, causing an estimated 450,000 infections annually [234,235]. This adverse impact has driven increased antibiotic use, which, in turn, has led to the emergence of antibiotic-resistant *Shigella* strains. These resistant strains pose a serious public health threat, as recognized by both the CDC and WHO [236]. An assessment of the antimicrobial resistance of *Shigella* in fresh fish samples imported into Jordan from Egypt, Yemen, and India showed that 98% of the isolates from the fresh fish samples displayed resistance to at least a single antimicrobial agent while 49% displayed resistance to at least three or more antimicrobial agents. The isolates exhibited high resistance to tetracycline, amoxicillin–clavulanic acid, cephalothin, streptomycin, and ampicillin [237]. The *Shigella* isolates from retail fish and shrimps from marine sources in Tanzania were resistant to tetracycline, gentamicin, and ciprofloxacin, while those from freshwater sources displayed no resistance to any antimicrobial agents [238].

*Listeria monocytogenes* can cause life-threatening diseases including meningitis, febrile gastroenteritis, encephalitis, spontaneous abortion, and septicemia [239]. Approximately 99% of all reported human listeriosis cases are foodborne and typically result from consuming foods such as raw milk and dairy, less-processed fish, meat, vegetables, ready-to-eat foods, and seafood [240,241]. About 7.2% of raw seafood contains 1–10^3^ CFU/serving, while less than 0.3% contains over 10^6^ CFU/serving [200]. Recent One Health reports from the EU indicate that *L. monocytogenes* is a significant etiologic agent in ready-to-eat fish and fishery products, with a prevalence rate of 7.1% in these products [242]. In Europe, the cases of listeriosis increased by 60% from 1381 cases in 2008 to 2206 cases in 2015 and were frequently reported among children, the elderly (≥65 years), and immunocompromised individuals [241]. In the USA, about 1600 people are infected and 260 people die each year due to *Listeria* infections [243]. Recent studies have shown that in seafood processing facilities, contamination with *L. monocytogenes* often arises from cross-contamination on food contact surfaces and equipment. It was reported *L. monocytogenes* was present in swabs from a fishing harbor (10.3%), fish landing centers (5.9%), and seafood processing plants (4.1%) [244]. Five clonal complexes of *L. monocytogenes* isolates from smoked herring, smoked salmon, and shrimp processing plants were also identified [245]. The study further reported that some of the isolates carried genes resistant to chemical disinfectants and biocides. Additionally, both persistent and sporadic strains of *L. monocytogenes* in a salmon filleting line, with the persistent strains showing greater tolerance to higher concentrations of most of the tested disinfectants, were also detected [246]. Food safety control measures have included synergistic combinations of interventions, such as sanitizing food contact surfaces, controlling storage temperatures and duration, and utilizing growth inhibitors and bacteriophages [239,247]. Due to the resilience and occurrence of *L. monocytogenes* in food processing environments, it serves as an indicator of hygiene and sanitation levels [248]. The antimicrobial resistance of *L. monocytogenes* in seafood has also been reported. For example, *L. monocytogenes* was prevalent in fresh and smoked salmon, as well as fresh cod in Poland [249]. The study further reported that some strains exhibited resistance to oxacillin (57.9%), ceftriaxone (31.6%), and clindamycin (8.8%), while two strains displayed multi-resistance to three classes of antimicrobials including cephalosporins, penicillins, and lincosamides. Also, *L. monocytogenes* isolates from seafood in Iran showed high resistance to ampicillin (100%), cefotaxime (100%), and penicillin (57%), but were sensitive to trimethoprim-sulfamethoxazole, chloramphenicol, and tetracycline [250]. Moreover, the antimicrobial susceptibility of *L. monocytogenes* isolated from salmon, shrimp, and tilapia imported into the USA from twelve countries indicated that *L. monocytogenes* isolates from all three seafood types exhibited 75% resistance to nitrofurantoin, with a minimum inhibitory concentration (MIC) of 128 µg/mL [251].

Another indicator for assessing sanitation and safety management practices in seafood processing environments is *Staphylococcus aureus* [252]. This bacterium harbors virulence factors such as Panton–Valentine leukocidin toxin (PVL), toxic shock syndrome toxin 1 (TSST-1), hemolysins, exfoliative toxins (ETs), and staphylococcal enterotoxins (SEs), which contribute to a plethora of human infections. These infections can be as mild as skin and soft tissue infections or as severe as septic arthritis, endocarditis, abscesses, necrotizing pneumonia, and osteomyelitis [253,254]. Among the virulence factors, SEs have a high tolerance to heat and cause food poisoning [200]. The prevalence of SEs in fish and fishery products varies widely, ranging from 2 to 60%. Concentrations as low as 0.02 ng/g of the toxins can cause food poisoning in susceptible individuals [255,256]. Infections from *S. aureus* are challenging to treat due to their high multi-antibiotic resistance. The WHO identified methicillin-resistant *S. aureus* (MRSA) as a high-priority antibiotic-resistant pathogen, requiring significant research [253]. An 8.75% occurrence of MRSA in raw mussels and cockles was observed, with complete resistance to amikacin and penicillin [257]. Additionally, 65.1% of *S. aureus* isolates from raw shrimps in China were multi-drug-resistant to antibiotics, including β-lactams, tetracycline, macrolides, aminoglycosides, lacosamide, streptogramin B, fosfomycin, trimethoprim, and streptothricin [255]. Furthermore, it was reported *S. aureus* shows resistance to penicillin (65.5%), tetracycline (22.5%), erythromycin (16.0%), and ciprofloxacin (3.2%) in tuna and salmon sashimi [258].

Fish and shellfish are among the vectors that transmit *Campylobacter* infections [174]. This bacterium resides in seafood that is contaminated with feces from seabirds such as gulls and shorebirds [259]. It causes gastroenteritis in humans, characterized by symptoms including abdominal pains, fever, bloody or watery diarrhea, cramps, and vomiting [174,260]. Campylobacter is the leading cause of bacterial diarrheal infections in the USA, with an estimated 1.5 million people getting ill from *Campylobacter* annually [261]. The trends in *Campylobacter* outbreaks in the USA from 1998 to 2016 showed that 8003 people became infected from 465 single-state outbreaks, averaging 24.47 outbreaks per year. *C. jejuni* was most commonly implicated, especially during the spring (27%) and summer (35%) seasons [262]. In the EU, campylobacteriosis is the most commonly reported foodborne illness with over 229,000 cases annually, resulting in economic and productivity losses of EUR 2.4 billion [263,264]. Its antimicrobial resistance in seafood has been reported. A reduced sensitivity to azithromycin and ciprofloxacin, which are key treatments for *Campylobacter* infections, resulted in approximately 36,800 infections per year in the USA in 2019 [265]. It was reported that *Campylobacter* spp. isolated from mussels in Spain exhibited multi-resistance to more than four antibiotics including ciprofloxacin, trimetroprim-sulphamethoxazole, ampicillin, and cefazolin [266]. Moreover, *Campylobacter* isolates from cockles, mussels, and oysters from various markets in Thailand displayed high resistance to erythromycin (72–84%), followed by nalidixic acid (28–40%) and ciprofloxacin (21–25%) [267].

##### Parasites as Pathogens in Seafood

Seafood-borne infections encompass a diverse array of pathogens including parasites, which collectively contribute to a wide range of clinical presentations. Parasitic diseases associated with seafood, however, remain significantly understudied compared to other types of infections. Currently, more than 40 parasite species have been documented in humans through seafood consumption. These parasites include various groups such as protozoa, cnidarians, tapeworms, flukes, roundworms, and acanthocephalans. The incidence of these infections is expected to rise with ongoing climate change, which can impact both the distribution and prevalence of parasites. Certain parasites, such as *Giardia*, *Toxoplasma*, *Clinostomum*, and *Anisakis*, have a global distribution and are frequently implicated in human disease. In contrast, species like *Corynosoma* are rarely reported in human hosts, while infections with *Anisakis* are documented regularly worldwide [152,268].

Seafood-borne parasites represent a highly diverse group in terms of morphology, life cycle, transmission dynamics, and clinical outcomes. This diversity contributes to significant variation in their migratory behavior within hosts and the clinical manifestations they induce. While a detailed examination of the life cycles and host interactions of these parasites exceeds the scope of this manuscript, an overview of the pathogenic potential and emerging significance of selected species is provided [269,270].

Parasites such as *Giardia* and *Toxoplasma* are ubiquitous and can be transmitted through contaminated water or undercooked seafood, posing a consistent public health concern. *Anisakis* species, which are nematodes found in various fish species, are notable for causing anisakiasis—a condition characterized by acute gastrointestinal symptoms following the ingestion of larvae. This disease is commonly reported in regions with a high consumption of raw or undercooked fish, emphasizing the connection between dietary habits and infection risk [152,271,272]. Conversely, infections with parasites like *Corynosoma* (belonging to the acanthocephalan group) are infrequent, reflecting the varied pathogenic potential within this category. Clinical presentations range from mild, self-limiting symptoms to severe, potentially life-threatening conditions, influenced by both the specific parasite and the host’s health status [152,273,274]. *Diphyllobothrium latum* (a cestode), commonly known as the broad fish tapeworm, causes diphyllobothriasis in humans through the consumption of raw or inadequately cooked freshwater fish, including marinated and smoked varieties [275,276]. Once ingested, this parasite colonizes the small intestine and upper colon, triggering symptoms such as indigestion, recurrent abdominal pain and distension, diarrhea, anorexia, muscular pain, numbness in the extremities, and the passing of tapeworm segments in feces [275]. *Paragonimus,* a trematode and lung fluke, causes pulmonary and extrapulmonary infections in humans, primarily through the consumption of second intermediate hosts, such as raw or inadequately cooked crabs and crayfish [277]. Similarly, the infective larvae of liver flukes, including *Clonorchis sinensis*, *Opisthorchis viverrini*, and *Fasciola hepatica*, in freshwater fish migrate to human bile ducts, maturing into adult flukes and subsequently causing inflammation. Chronic infections with these parasites may lead to severe health conditions, such as cholangiocarcinoma, various hepatobiliary disorders, and fascioliasis [277]. *Gnathostoma* (a nematode) has an infective phase during its third-stage larval form, which develops in fish and other marine organisms, including mollusks. Upon the ingestion of the infected fish, the larvae migrate through various organs and tissues, causing gnathostomiasis. Symptoms can range from mild skin inflammation to severe complications involving the central nervous system and vision [278].

The morphological and ecological diversity of these parasites is mirrored in their complex life cycles and host interactions. For instance, *Anisakis* spp. undergo multiple developmental stages involving intermediate hosts in marine ecosystems before maturing in definitive hosts such as marine mammals. When humans consume infected seafood, they become accidental hosts, disrupting the parasite’s life cycle and triggering an inflammatory response as the larvae migrate through the gastrointestinal tract. These migratory behaviors and host adaptations contribute to distinct clinical syndromes [279,280].

Given the growing incidence of seafood-borne parasitic infections—potentially exacerbated by climate change and global seafood trade—enhanced research, monitoring, and public health measures are crucial. The emergence of these parasites as public health threats calls for improved surveillance and understanding of their pathogenicity to inform strategies that mitigate the risks associated with seafood consumption [152,273].

##### Harmful Algal Blooms and Their Biotoxins as Pathogens in Seafood

Algae in marine environments can proliferate and accumulate on the surface of waters under favorable conditions of nutrients, climate, and hydrology, forming clusters known as Harmful Algal Blooms (HABs) [281]. These blooms can have devastating effects on marine ecology by reducing water quality, disrupting ecosystems, increasing mortality among aquatic animals, and causing hypoxia and anoxia due to the high concentrations of algae. Additionally, they produce biotoxins that pose significant risks to seafood safety and human health [281,282]. HABs severely impact the fish and shellfish industry, affecting aquaculture, coastal tourism, recreational fisheries, and wildlife. These effects have led to the development of monitoring programs like GLOBALHAB Status, with the USA being one of the predominant locations for monitoring [283,284]. For example, a domoic acid event in 2015 on the USA West Coast caused a direct loss of USD 48.3 million from a delayed Dungeness crab season, resulting in USD 25 million in disaster aid from Congress [285]. Furthermore, an *Alexandrium catenella* bloom in 2005 caused significant economic losses in Maine and Massachusetts [283]. Recently, the CDC reported 368 HAB events across 16 states in 2021, resulting in 117 human illnesses and at least 2715 animal illnesses [286]. The rapid increase in HABs worldwide has been linked to eutrophication, climate change, increased aquaculture, frequent coastal travel, and urban growth [287,288]. Common HABs include *Pseudo-nitzschia*, *Alexandrium*, *Dinophysis*, *Gymnodium*, *Karenia*, *Karlonidium*, *Prorocentrum*, and *Heterosigma* [287,289].

Harmful algal blooms (HABs) produce biotoxins, which include secondary metabolites and small molecules such as oligopeptides, alkaloids, and lipopolysaccharides. These toxins disrupt major metabolic processes and cause significant health risks to both humans and animals [290]. Biotoxins such as okadaic acid, saxitoxin, domoic acid, azaspiracid, and brevetoxins, which are primarily found in shellfish, cause Diarrhetic Shellfish Poisoning (DSP), Paralytic Shellfish Poisoning (PSP), Amnesic Shellfish Poisoning (ASP), Azaspiracid Shellfish Poisoning (AZP), and Neurotoxic Shellfish Poisoning (NSP), respectively, in humans [291]. These biotoxins cause significant global health risks, with more than 20,000 reported cases of food poisoning attributed to marine biotoxins each year [292]. For example, Ciguatera poisoning occurs at a rate of 251 cases per 10,000 persons each year among many Pacific and Caribbean populations [293]. Due to growing concerns about marine biotoxin poisoning, rapid testing methods such as biosensors, antibody-based lateral flow assays, ELISA, receptor binding assays, HPLC-UV, LC-MS/MS, HPLC-FLD, and mouse bioassays have been implemented to detect these biotoxins in seafood [292,294]. Moreover, programs such as GLOBALHAB and the national human shellfish poisoning surveillance program in France have been established to address issues related to HABs [283,295]. In Brazil, the National Program for Hygienic-Sanitary Control of Bivalve Mollusks (PNCMB) was established in 2012 to set limits for the bacterial contaminants and biotoxins originating from HABs in shellfish-producing areas [296]. Similarly, the European Union (EU), through Regulation (EC) No 853/2004 and Regulation (EC) No 854/2004, requires member states to implement monitoring programs to ensure harvested shellfish from classified production areas comply with the established maximum levels for marine biotoxins and adhere to specific hygiene procedures [297].

**Table 2 foods-14-01461-t002:** Occurrence of foodborne pathogens in seafood and their harvesting sites.

Pathogen		Source	Country	Reference
Genus	Genotypes/Species/Strains/Serovars			
Viruses				
Norovirus (NoV)	HuNoV Group II (GII)	Oysters, clams, shrimps, and finfish	India	[298]
	Norovirus GI and GII	Sea urchins	Portugal	[299]
	Norovirus GI and GII	Oysters	USA	[300]
Hepatitis A (HAV)	N.D.	Fish and shrimps	Iran	[301]
	N.D.	Oysters	USA	[300]
	Genotype IA	Mussels and clams	Italy	[302]
Hepatitis E (HEV)	G3 HEV	Mussels and clams	Italy	[303]
	G3 HEV	Mussels and oysters	Scotland	[304]
	4 Sub-genotypes (4b/4d)	*A. granosa*, *S. subcrenata*, and *R. philippinarum*	China	[305]
Adenovirus (AdV)	N.D.	Finfish, bivalve mollusks, crustaceans, and cephalopods	India	[306]
	A species HAdVs serotype 12, F species HAdVs serotype 41, and C species PAdVs serotype 5	Oysters	Taiwan	[307]
	HAdV	Mussels	Spain, Greece, and Finland	[308]
Sapovirus (SaV)	N.D.	Clams, cockles, and oysters	Morocco	[309]
	N.D.	Shrimp, oysters, Atlantic salmon, and Arctic surf clams	China	[310]
	GI.1, GI.2, GI.3, GII.4, and GV.1	Mussels	Spain	[311]
Rotavirus (RV)	N.D.	Mussels and clams	Italy	[302]
	N.D.	Mussels and clams	India	[312]
	G1, G2, G3, and G9	Oysters	Thailand	[313]
Aichivirus (AiV)	AiV-1	Oysters and clams	Vietnam	[314]
	AiV-1, genotype B	Mussels	South Africa	[315]
	AiV-1, genotype B	Mussels, oysters, and clams	Italy	[316]
Bacteria				
*Vibrio*	*V. parahaemolyticus*	Oysters	USA	[317]
	*V. parahaemolyticus*, *V. vulnificus*, *V. alginolyticus*, and *V. mimicus*	Oysters, clams, mussels, fish, and shrimp	Mexico	[172]
	*V. parahaemolyticus*, *V. vulnificus*, and *V. cholerae*	Mussels, clams, oysters, fish, cephalopods, and crustaceans	Italy	[318]
*Shewanella*	*S. algae*	Oysters, clams, and abalone	Taiwan	[319]
	*S. algae*	Shrimps	India	[320]
	*S. abalonesis*, *S. algae*, *S. baltica*, *S. hafniensis*, *S. marisflavi*, and *S. putrefaciens*	Oysters	USA	[321]
*Pseudomonas*	*P. aeruginosa*	Fish (fresh, smoked, salted dried, and frozen)	Iran	[322]
	*P. fluorescens*, *P. fragi*, *P. lundensis*, *P. marginalis*, *P. syringae*, *P. taetrolens*, *P. chlororaphis*, *P. tolaasii*, and *P. viridilivida*	Salmon, plaice, and northern cod fillets	Italy	[191]
	*P. brenneri*, *P. defensor*, *P. haemolytica*, *P. lactis*, *P. lundensis*, *P. lurida*, *P. mandelii*, *P. meridiana*, *P. migulae*, *P. proteolytica*, *P. simiae*, and *P. weihenstephanensis*,	Rainbow, Black Sea, brook, and brown trout	Turkey	[323]
*Clostridium*	*C. botulinum*	Canned tuna, sardine, and mackerel	Egypt	[201]
	*C. perfringes*	Oysters, mud snails, scallops, clams, loach, and *monopteros albus*	China	[181]
	*C. difficile*	Oysters	USA	[324]
*Escherichia coli*	*E. coli*	Shrimp, catfish, and tilapia	USA	[174]
	β-lactamase and carbapenemase-producing *E. coli*	Clams, oysters, razor clams, cockles, and mussels	Portugal	[325]
	Shiga-toxin-producing *E. coli* (STEC)	Shrimps, crabs, and oysters	Egypt	[220]
*Salmonella*	*S. Enterica* subsp. *enterica Typhimurium* and monophasic variant I 1;4, [5], and 12:i:-	Oysters, mussels, and clams	Canada	[326]
	*S. Typhimurium*, *S. Enteritidis*, *S. Branderup*, *S. Infantis*, *S. Virchow*, *S. Agona*, and *S. Derby*	Clams, mussels, oysters, and scallops	Poland	[225]
	*S. Typhimurium*	Shrimp, catfish, and tilapia	USA	[174]
*Shigella*	*Shigella* spp.	Red mullet, red sea bream, and emperor fish	Jordan	[237]
	*Shigella* spp.	Salmon, shrimp, and tilapia	USA	[251]
	*S. dysenteriae* and *S. flexneri*	Tilapia, common carp, and catfish	Ethiopia	[327]
*Listeria*	*L. monocytogenes*	Smoked seafood, seafood salads, and fresh crab meat or sushi	USA	[328]
	*L. monocytogenes*	Bogue, horse mackerel, hake, chub mackerel, European anchovy, and European pilchard	Greece	[242]
	*Listeria* spp. and *L. monocytogenes*	Fish products, sushi, and other ready-to-eat foods	Poland	[248]
*Staphylococcus*	*S. aureus*	Raw processed fish and other ready-to-eat foods	Bangladesh	[253]
	*S. aureus*, *S. xylosus*, *S. lugdunensis*, *S. hominis*, *S. haemolyticus*, *S. lentus, S. sciuri*, and *S. capitis*	Dried, smoked, and braised fish	Burkina Faso	[329]
	*S. aureus*	Freshwater fish, saltwater fish, and shrimp	China	[254]
*Campylobacter*	*C. jejuni*	Shrimp, catfish, and tilapia	USA	[174]
	*C. lari* and *C. jejuni*	Mussels	Croatia	[259]
	*C. lari*, *C. jejuni*, *C. lari*, and *C. peloridis*	Oysters, mussels, and common cockle	France	[264]
Algae blooms				
*Alexandrium*	*A. Pacificum* and *A. minutum*	Shellfish and water	Italy	[330]
	*A. catenella*, *A. foedum*, *A. insuetum*, *A. lee*, *A. margalefi*, *A. minutum*, *A. pseudogonyaulax*, and *A. tamarense*	Clam and water	Tunisia	[331]
	*A. tamarense*	Seawater	Scotland	[332]
*Dinophysis*	*D. acuminata*, *D. norvegica*, *D. fortii*, *D. ovum*, and *D. caudata*	Water	USA	[333]
	*D.* cf. *acuminata*	Water	Greece	[334]
	*D. acuminata*, *D. norvegica*, and *D. acuta*	Water	Norway	[335]
*Gymnodium*	*Gymnodinum* spp.	Seawater	Morocco	[336]
	*G. catenatum*	Water	Portugal	[337]
*Karlodinium*	*K. australe*	Water	Malaysia	[338]
	*K. veneficum*	Water	China	[339]
	*K. veneficum*	Water	China	[340]
*Prorocentrum*	*P. minimum*	Water	Mexico	[341]
	*P. lima* complex, *P. caipirignum*, *P*. cf. *concavum*, *P*. cf. *emarginatum*, *P*. cf. *fukuyoi*, and/or *P*. cf. *rhathymum*	Water	Japan	[342]
	*P. lima* complex, *P. rhathymum*, *P. borbonicum*, *P. levis*, *P. rhathymum*, and *P. emarginatum*	Water	Greece	[343]

N.D.: not determined.

**Table 3 foods-14-01461-t003:** Outbreaks of foodborne pathogens from seafood within the last decade (2014–2024).

Pathogen	Contaminated Seafood	Year	Facts	Symptoms	Country/Continent	Reference
*Listeria monocytogenes* ST173	Fish products	2012–2024	Multi-country (7); prolonged outbreak; 73 cases; 14 deaths	N.R.	Europe	[344]
*Listeria monocytogenes* ST1607	Smoked salmon products	2021–2024	Multi-country (3); prolonged outbreak; 20 cases; 5 deaths	N.R.	Europe	[345]
*Listeria monocytogenes* ST155	Ready-to-eat fish products	2016–2023	Multi-country (5); prolonged outbreak; 64 cases; 10 deaths	N.R.	Europe	[346]
Norovirus	Raw oysters	2022	Multistate (8); 211 illnesses	Fever, nausea, diarrhea, vomiting, abdominal cramps, chills, and headache	USA	[347]
Norovirus	Raw oysters	2022	Multistate (13); 192 illnesses	Diarrhea, vomiting, nausea, fever, headache, body ache, and stomach pain	USA	[347]
*Salmonella* Litchfield	Fresh fish	2022	Multistate (4); 39 illnesses; 15 hospitalizations; 0 deaths	N.R.	USA	[348]
*Vibrio parahaemolyticus* ST417 and ST50	Raw oysters	2021–2022	184 confirmed cases; 27 hospitalizations; 0 deaths	Diarrhea and abdominal pain	Australia	[349]
*Salmonella* Thompson	Seafood	2021	Multistate (15); 115 illnesses; 20 hospitalizations; 0 deaths	N.R.	USA	[350]
*Salmonella* Weltevreden	Frozen cooked shrimp	2021	Multistate (4); 9 illnesses; 3 hospitalizations; 0 deaths	N.R.	USA	[351]
N.I.	Mixed spicy seafood salad	2020	368 cases; 0 deaths	Abdominal pain, watery diarrhea, nausea, vomiting, fever, and bloody diarrhea	Thailand	[352]
Hepatitis A	Clams, snapping shrimps, cockle, corbicula, and oysters	2020	191 cases	N.R.	China	[353]
*Vibrio parahaemolyticus*, *Shigella flexneri*, Shiga toxin-producing *E. coli* non-O157, *Vibrio albensis*, *Campylobacter lari*, and norovirus genogroup 1.	Raw oysters	2019	Multistate (5); 16 illnesses/cases; 2 hospitalizations; 0 deaths	Diarrhea (that may be watery or bloody), stomach cramps or pain, nausea, vomiting, and fever	USA	[354]
*Salmonella* Newport	Frozen raw tuna	2019	Multistate (8); 15 illnesses/cases; 2 hospitalizations; 0 deaths	Diarrhea, fever, and stomach cramps	USA	[355]
*Vibrio mimicus ctx*-negative	Steamed blue crab, snow crab, shrimp, and oysters	2019	One state (Florida); 6 cases; 4 hospitalizations; 1 intensive care	Vomiting, headache, and nausea	USA	[356]
*Listeria monocytogenes* ST1247 (clonal complex 8)	Cold-smoked fish products	2014–2019	Multi-country (5); prolonged outbreak; 22 cases; 5 deaths	N.R.	Europe	[357]
*Vibrio parahaemolyticus*	Fresh crab meat	2018	8 jurisdictions; 26 illnesses/cases; 9 hospitalizations; 0 deaths	N.R.	USA	[358]
*Listeria monocytogenes* ST8	Salmon products	2015–2018	Multi-country (3); prolonged outbreak; 12 cases; 4 deaths	N.R.	Europe	[359]
Hepatitis A virus	Frozen scallops	2016	1 state (Hawaii), 74 hospitalizations	Yellow eyes or skin, abdominal pain, pale stools, and dark urine	USA	[360]
*Vibrio cholerae*	Raw seafood	2016	3 cases	Watery diarrhea, anorexia, vomiting, yellowish diarrhea, abdominal pain, myalgia, and acute renal failure	South Korea	[361]
*Salmonella* Paratyphi B variant L(+) tartrate(+) and Weltevreden	Frozen raw tuna	2015	Multistate (11); 65 illnesses/cases; 11 hospitalizations; 0 deaths	N.R.	USA	[362]
N.I. (suspected shellfish poisoning)	Clams	2015	20 cases (January); 199 cases (April)	Dizziness, vomiting, nausea, headache, abdominal cramps, diarrhea, fever, and perioral and distal paresthesia	India	[363]

N.R.: not reported; and N.I.: not identified.

## 5. Future Perspectives: Strategies to Control the Risks Associated with Seafood Consumption

In light of the various health hazards associated with seafood consumption discussed in this review, there is an increasing need to implement robust strategies to reduce the associated risks, particularly amid the projected rise in global seafood consumption. These interventions should adopt a synergistic approach that requires coordinated efforts across aquaculture production, harvesting, post-harvest processing, storage, supply chain management, regulatory control, and consumption (Figure 4) [364].

The enhanced and continuous surveillance of seafood hazards is an effective strategy that could be used to mitigate the risks associated with seafood consumption. Effective monitoring systems, as part of national and global surveillance programs, facilitate the rapid identification of both known and emerging pathogens, validate the safety of harvesting sites, inform appropriate post-harvest processing requirements, and reveal long-term contamination trends, ultimately helping to reduce food safety risks [365]. Currently, systems such as the Foodborne Diseases Active Surveillance Network (FoodNet), the National Outbreak Reporting System (NORS), and the Foodborne Disease Outbreak Surveillance System (FDOSS) in the United States; the China National Center for Food Safety Risk Assessment (CFSA); and the European Surveillance System (TESSy) in Europe are focused on tracking emerging pathogens and chemical contaminants, identifying outbreak patterns, and enabling timely responses to public health threats [366,367,368]. Traditional microbiological methods, including culturing and biochemical assays, have long-served as the cornerstone for food safety monitoring by allowing the detection and quantification of pathogens in various food products [369,370]. Molecular biology techniques, such as the polymerase chain reaction (PCR), its quantitative variant (qPCR), loop-mediated isothermal amplification (LAMP), and next-generation sequencing (NGS), offer substantial advantages over traditional microbiological methods in the monitoring of pathogens. Through enhanced sensitivity, specificity, and speed, these techniques facilitate the rapid and accurate detection of foodborne pathogens, making molecular biology an essential tool in ensuring the safety of seafood products [7,8,371,372,373,374,375]. These systems and techniques can be improved for enhanced monitoring by incorporating advanced and novel digital methods, such as the Internet of things (IoT), big data, whole-genome sequencing (WGS), blockchain, text mining, machine learning, and artificial intelligence (AI), to ensure real time warning and the identification of threats to seafood safety [376]. For example, digital technologies can support the development of predictive monitoring indicators for weather anomalies that signal risks of pathogen emergence in aquaculture environments, as runoff from extreme weather events can increase the likelihood of contamination. Moreover, aquaculture producers and post-harvest processors can employ digital tools such as radiofrequency identification (RFID) tags and readers, GPS tracking systems, and biosensors for the early detection, semi-quantification, and quantification of foodborne pathogens, toxins, and chemical contaminants [376]. Additionally, AI-powered data analytics can be used to predict contamination risks based on environmental and supply chain factors, facilitating proactive rather than reactive responses to contamination risks [377].

Seafood traceability is an effective tool for managing risks across the seafood supply chain. It enables the tracking of seafood products through all stages: from harvesting and production to processing, distribution, and final consumption [378]. Traceability ensures the transmission of relevant information throughout the supply chain, thereby enhancing food safety, quality assurance, and control mechanisms and facilitating efficient recalls of contaminated products to minimize public health risks and economic losses [379]. One of the major benefits of seafood traceability is its ability to combat seafood fraud, which poses serious risks to public health. Seafood fraud refers to the intentional misrepresentation and mislabeling of seafood products, typically for economic gain, by providing false or misleading information to consumers. This includes species substitution; fishery substitution; species adulteration; illegal, unreported, and unregulated (IUU) substitution; undeclared product extension; catch method fraud; chain-of-custody abuse; modern-day slavery; and animal welfare [380]. These practices pose significant public health risks, as they can expose consumers to seafood products with undeclared allergens, unapproved or unsafe products, allergens, toxins, and undeclared antibiotics [380]. Robust traceability systems including documentation, auditing, testing, and regulatory enforcement are important for mitigating seafood fraud and its associated health risks [380,381]. Emerging digital tools, such as blockchain technology, offer significant advantages for seafood traceability by ensuring immutability, data integrity, transparency, and enhanced supply chain visibility, to facilitate faster recalls, reduce counterfeit products, and support safer seafood trade [382]. Moreover, vessel monitoring systems (VMSs), key data elements, electronic logbooks, and electronic monitoring systems can improve traceability efforts and ensure the quality and safety of seafood products [381].

Given the rising global demand for seafood, adopting sustainable aquaculture practices is relevant not only for environmental conservation but also for promoting public health. Sustainable aquaculture practices, including improving biosecurity systems and minimizing the use of antibiotics, is key to minimizing the risks associated with seafood consumption [383]. Biosecurity in aquaculture involves the effective management of pathogen-related risks throughout the production cycle. This includes sourcing healthy stocks, optimizing husbandry practices, disinfecting facilities, implementing disease detection and diagnostic systems, engaging stakeholders across the supply chain, and adhering to appropriate policy frameworks [384,385]. Within production facilities, effective biosecurity measures prevent pathogen introduction and spread through sanitation protocols and ensure regular surveillance and the administration of probiotics to support animal health. However, the widespread and often unregulated use of antimicrobial agents in aquaculture poses significant public health concerns [385]. The overuse of antibiotics can result in drug residues in seafood, which contribute to antimicrobial resistance and increase the risk of hypersensitivity reactions; toxicity; and carcinogenic, mutagenic, and teratogenic effects in consumers [386]. Although antimicrobial agents have relevant uses in aquaculture production, minimizing and regulating their use is an effective way to reduce the associated risks in seafood consumption [386].

Harmonizing and enforcing regulations governing seafood safety provides a solid framework for ensuring consistency, accountability, and compliance throughout the global seafood supply chain. The absence of uniform global standards presents considerable challenges in enforcing international food safety measures, highlighting the need for regulatory alignment to promote a more unified and standardized approach [387]. Currently, many seafood regulations are developed to address specific challenges and opportunities unique to individual jurisdictions, which can complicate international seafood trade. A globally harmonized regulatory framework helps bridge the disparities created by these differing national policies, thereby supporting collective action against seafood-associated risks, particularly in imported products [387]. Additionally, the strict enforcement of current seafood regulations, including those addressing seafood fraud, labeling, antimicrobial agents, and sustainable aquaculture practices, is important in minimizing the risks associated with seafood consumption [380,385,386,388]. In countries such as Norway, Canada, Ireland, the USA, New Zealand, and the UK, the Hazard analysis and critical control points (HACCP) system controlled by regulatory authorities is widely implemented across the seafood supply chain to ensure product safety. Moreover, the application of HACCP at the farm level has been proposed as a measure to address concerns surrounding food hygiene, safety, and authenticity [386]. Within aquaculture and seafood processing facilities, strict adherence to HACCP-based systems, including supplier verification, sourcing from approved harvesting areas, regular testing, hygiene protocols, and proper handling and storage, can effectively reduce the risks of pathogens, histamine, and chemical contaminants in seafood. Ultimately, regardless of how comprehensive regulations and inspections may be, the final layer of protection lies with the consumer. Through proper handling and preparation, seafood consumers can help maintain the safety of seafood and reduce the risks associated with its consumption [389].

## 6. Conclusions

Seafood plays a vital role in global nutrition, providing high-quality proteins, essential fatty acids (e.g., EPA and DHA), and key micronutrients, such as zinc, iodine, and vitamin D. These nutrients are essential for maintaining human health, supporting cardiovascular function, promoting optimal neurodevelopment, and modulating inflammatory processes. However, the benefits of seafood consumption must be carefully considered based on the potential risks associated with both natural processes and human activities. Unfortunately, the contamination of the marine ecosystem introduces a range of chemical hazards into seafood. Heavy metals such as mercury and cadmium, which bioaccumulate in higher trophic species, pose serious health risks, including neurotoxicity and kidney damage. Persistent organic pollutants act as endocrine disruptors and are linked to carcinogenic effects. Additionally, the widespread presence of microplastics and their toxic chemical additives in seafood has become an emerging concern, with significant impacts on human health and marine biodiversity. To address these challenges, a comprehensive, multi-disciplinary approach is essential. Strengthening national and global food safety monitoring systems enables the early detection of emerging pathogens and contaminants, while digital technologies such as the IoT, AI, and whole-genome sequencing further enhance predictive capabilities and rapid response. Robust seafood traceability mechanisms and enforcing seafood regulations across the entire supply chain can facilitate timely recalls, combat seafood fraud, and improve transparency, ultimately reducing public health risks and economic losses. Collaboration across fields, including marine biology, environmental science, food safety, and public health, is crucial for balancing the nutritional benefits of seafood with its associated risks. Most importantly, policymakers must prioritize the development of regulatory frameworks and support research in the field of emerging contaminants and pathogens. By integrating scientific innovation and fostering environmental responsibility, the seafood industry can ensure that this vital resource remains safe, sustainable, and accessible for the next generations.

## Figures and Tables

**Figure 1 foods-14-01461-f001:**
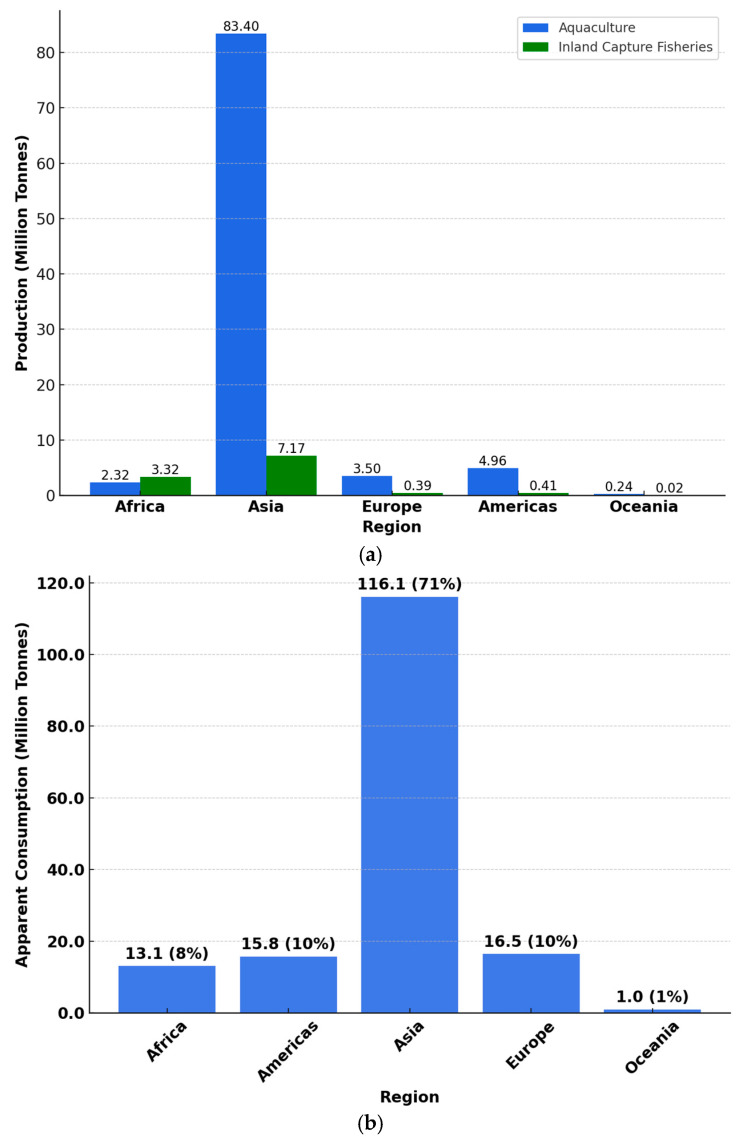
Regional distribution of (**a**) aquaculture and inland capture fisheries production (2022); and (**b**) apparent consumption of aquatic animal foods (2021). Data were retrieved from [16].

**Figure 2 foods-14-01461-f002:**
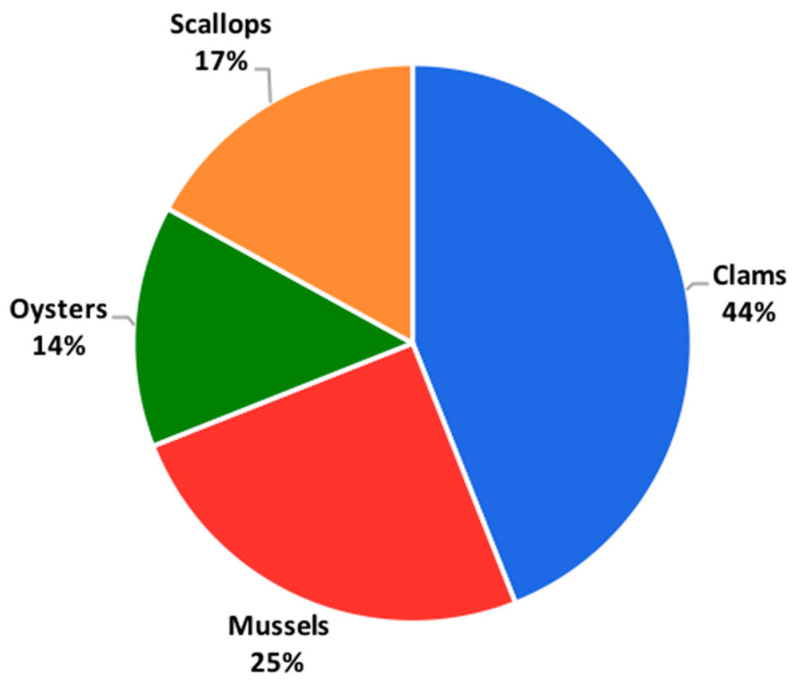
Estimates of dietary exposure to microplastics from the consumption of different mollusks (the percentages were calculated based on the data retrieved from [66]).

**Figure 3 foods-14-01461-f003:**
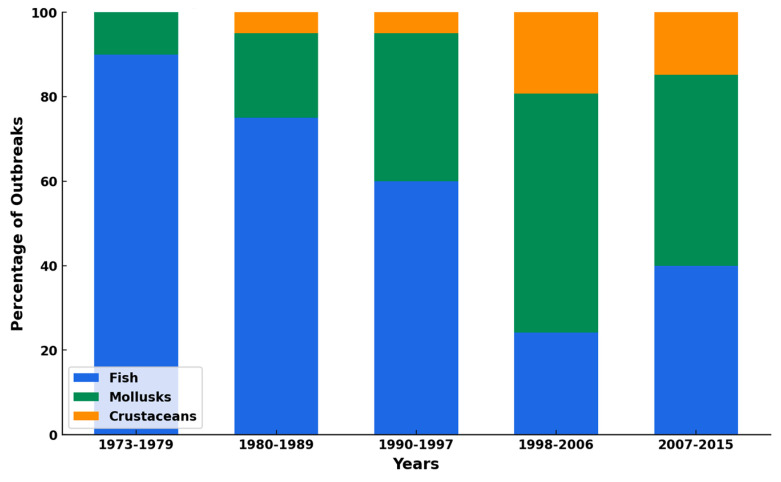
Seafood-associated outbreaks (%) attributable to seafood commodities within five multi-year ranges (data were retrieved from [153,154]).

**Figure 4 foods-14-01461-f004:**
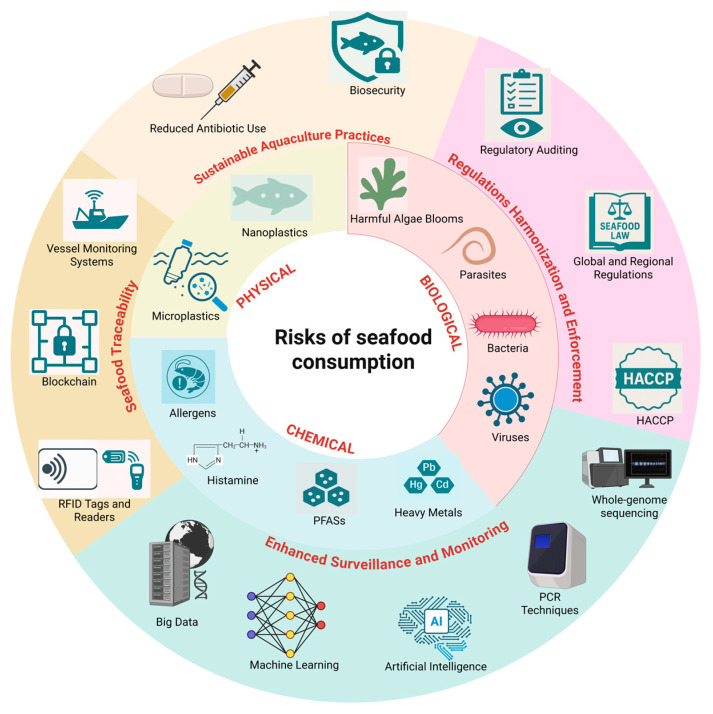
Multifaceted strategies for controlling risks associated with seafood consumption.

## Data Availability

The original contributions presented in the study are included in the article, further inquiries can be directed to the corresponding author.

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
