# Peer review of "Exploring the Complexities of Seafood: From Benefits to Contaminants"

_foods, 2025, doi:10.3390/foods14091461_

Round 1

Reviewer 1 Report

Comments and Suggestions for Authors

The work compiles important data regarding the risks associated with the consumption of seafood. However, I made several suggestions to meet the scope of the journal, improving its technical-scientific quality. 

Please see the detail comment in the attachment.

Reviewer 2 Report

Comments and Suggestions for Authors

The review submitted by Taylor et al. provides an exceptionally in-depth analysis of the nutritional aspects of fishery products, as well as the biological, chemical, and physical hazards associated with this category of food.

The topic is highly relevant, and the review is both comprehensive and well-developed, with an extensive number of references and a detailed writing style.

However, I have two main suggestions:

  1. The data presented are often outdated. For instance, when discussing the occurrence of biological hazards in these foods, the authors rely on data that are more than ten years old, which necessitates complete revision and update (see Figure 5, for example). The authors should incorporate more recent data on the issue discussed.
  2. The authors fail to integrate the concept of molecular biology with the rest of the review. This section appears as a brief and somewhat disconnected topic within the manuscript. There seems to be no clear rationale for including this topic in the review, and at the same time, the authors do not provide solutions to the extensively described biological, chemical, and physical hazards. I suggest that the authors remove the entire molecular biology section and, instead, include a section outlining the key preventive measures for the identified hazards, along with “the way forward.”

Others specific Revisions

  • L68: Use italics for bacterial genera.
  • L178, 232, 531: Provide references for the cited documents.
  • L369, 831: The term "vector" is not appropriate, as vectors are invertebrate animals. Using "transmitters" may be more epidemiologically accurate.
  • L530: Provide more details regarding the outbreaks. Where did they occur? Are these global data or specific to a particular region?
  • Figure 5: Are there no more recent data available? The latest data included are from 2006, making them nearly 20 years old. Presenting this information in its current format is not meaningful. The authors should include up-to-date data on this topic for the readers.
  • L542: Update.
  • L576: Update.
  • Section 4.4.1: Including some recent outbreaks would be valuable.
  • L592: Standardize the capitalization of "Gram," ensuring it begins with an uppercase letter throughout the manuscript.

Reviewer 3 Report

Comments and Suggestions for Authors

Dear Editor,

I would like to thank the authors for providing important information regarding seafood and especially for covering in a very profound way all the aspects from benefits to contaminants and pathogens.

The manuscript is well written and adequately cited. All Tables and Figures are well designed, facilitating even less experienced readers to understand better the scope of the paper.

I have no major comments regarding this manuscript which is of publishable quality.

I have only a few minor comments/suggestions for the authors’ consideration.

  1. In Part 4 of the manuscript, harms have been haphazardly placed in the text without following certain criteria. Probably, a separation to chemical and biological hazards could be more helpful for the readers.  
  2. Line 531. No reference

Round 2

Reviewer 2 Report

Comments and Suggestions for Authors

All my suggestions were adressed.